# Learning Fast and Accurate Machine Learning Force Fields via Joint Atomic Energy and Energy Hessian Distillation

## Abstract

Atomistic foundation models, trained on extensive and diverse datasets, now achieve near *ab initio* accuracy across broad molecular and material systems while demonstrating strong transferability across chemical spaces. However, their large parameter counts result in high inference latency and large memory requirements, hindering long-time-scale molecular dynamics simulations and deployment on resource-constrained hardware. In practice, researchers in physical chemistry often focus on specific chemical subdomains, where compact specialized models with fewer parameters would be sufficient—provided they inherit appropriate inductive biases from large foundation models. This need motivates distillation techniques that compress foundation models into efficient specialized models while preserving accuracy. In this paper, we propose an architecture-agnostic distillation method: Joint Atomic Energy–Energy Hessian Distillation. This approach augments state-of-the-art Hessian supervision with atomic energy, which complements low-frequency components at minimal computational overhead ($<0.5\%$). Compared with the current state-of-the-art method, our method consistently improves energy MAE over Hessian-only distillation (averaging 48.3% on SPICE and 6.1% on MPtrj datasets) while achieving comparable force MAE (average improvement of 1.4%). Ultimately, our approach reduces parameter counts by 78%–98%, enabling fast and deployment-friendly specialized models for targeted chemical subdomains.

## 1 Introduction

Foundation models (FMs) have emerged as a powerful tool in computational materials science, demonstrating remarkable accuracy and generalization capabilities in property prediction and materials discovery (Deng et al., 2023; Batatia et al., 2023; Fu et al., 2025; Kovács et al., 2025; Wood et al., 2025). These gains stem from large, heterogeneous quantum-based datasets spanning molecules and materials, including OC20/OC22 for catalysis (Chanussot et al., 2021; Tran et al., 2023), SPICE/OMol25 for molecules (Eastman et al., 2023; 2024; Levine et al., 2025), and MPtrj/OMat24 for materials (Deng et al., 2023; Barroso-Luque et al., 2024), as well as architectures that combine message passing with strong physical inductive biases such as invariance and equivariance to capture complex interatomic interactions.

Despite this progress, the architectural complexity and large parameter counts of FMs limit their practicality for million-step molecular dynamics and large-scale relaxations. specialized machine learning force fields (MLFFs) such as DeePMD (Wang et al., 2018), PaiNN (Schütt et al., 2021), and GemNet (Gasteiger et al., 2021) provide much faster inference and can support billion-atom simulations at nanoseconds per day on top supercomputers (Jia et al., 2020; Guo et al., 2022). In many studies, researchers focus on specific chemical subdomains, such as specific elements, space groups, or biomolecular families. In such settings, compact and fast models are sufficient (Unke et al., 2021). This motivates transferring the capabilities of FMs into small specialized models.

Knowledge distillation (KD) (Hinton et al., 2015; Gou et al., 2021) is a well-established method for improving the speed-accuracy trade-off by transferring information from a large teacher model to a smaller student model. Beyond classic logit-based KD, feature-based protocols align intermediate

representations (e.g., node, edge, or vector features) and have recently been adapted to molecular graph neural networks (GNNs), yielding accuracy gains in energy/force regression without changing student architectures or reducing throughput (Das et al., 2023; Ekström Kelvinius et al., 2023; Sheshanarayana & You, 2025). However, feature matching can be brittle when teacher and student differ in inductive biases and internal feature parameterizations.

A complementary approach is to distill architecture-agnostic and physically meaningful information directly from the energy surface itself. Recently, Amin et al. (2025) proposed the current state-of-the-art Hessian distillation method, which aligns rows of the teacher's energy Hessians with those of the student during training. This Hessian distillation transfers local curvature (Rodriguez et al., 2025), while remaining agnostic to internal feature choices and working across directional/equivariant designs and across direct-force/conservative-force parameterizations. Conceptually, training to match function derivatives echoes the broader idea of Sobolev training (Czarnecki et al., 2017), which can improve sample efficiency and generalization by supervising gradients or higher-order derivatives.

In this paper, our goal is to distill fast, domain-specialized small models from large foundation models for specific chemical subdomains such as selected element families, space groups, or molecular families while preserving throughput, deployment friendliness, and consistent energy and force accuracy. We first present a spectral analysis of Hessian distillation, formalizing that the errors of energy, forces, and Hessians share the same Fourier coefficients weighted by 1, $\omega^2$, and $\omega^4$, respectively. Building on this insight, we propose joint Atomic Energy–Energy Hessian distillation. Atomic energy decomposition is a commonly used method in MLFFs and can complement low-frequency components without incurring much overhead. We demonstrate our method on the foundation models MACE-OFF (Kovács et al., 2025) trained on SPICE (Eastman et al., 2023; 2024), MACE-MP (Batatia et al., 2023) trained on MPtrj (Deng et al., 2023), and eSEN (Fu et al., 2025) trained on MPtrj, sAlex, and OMat24 (Barroso-Luque et al., 2024), where the joint objective consistently outperforms Hessian distillation on energy MAE (averaging 48.3% on SPICE and 6.1% on MPtrj) with minimal computational overhead ($<0.5\%$), while achieving comparable force MAE to Hessian distillation (average improvement of 1.4%). Ultimately, our method delivers a 78%–98% reduction in parameter counts, enabling fast and deployment-friendly specialized models for targeted chemical subdomains.

## 2 PRELIMINARIES

**Machine Learning Force Fields.** Given a system of $N$ atoms with Cartesian coordinates $\boldsymbol{R} = (\boldsymbol{r}^{(1)}, \ldots, \boldsymbol{r}^{(N)}) \in \mathbb{R}^{N \times 3}$ and atomic numbers $\boldsymbol{Z} = (z^{(1)}, \ldots, z^{(N)}) \in \mathbb{R}^N$, a MLFF predicts the total potential energy $\hat{E}_{\text{tot}} \in \mathbb{R}$ and per-atom forces $\hat{\boldsymbol{F}} = (\boldsymbol{f}^{(1)}, \ldots, \boldsymbol{f}^{(N)}) \in \mathbb{R}^{N \times 3}$. Typically, the total energy is parameterized via an atomic decomposition. The model first outputs atomic energies $\hat{\boldsymbol{E}}_{\text{atom}} = (\hat{E}_{\text{atom}}^{(1)}, \ldots, \hat{E}_{\text{atom}}^{(N)}) \in \mathbb{R}^N$, and aggregates them as:

$$\hat{E}_{\text{tot}} = \sum_{i=1}^{N} \hat{E}_{\text{atom}}^{(i)}. \tag{1}$$

With reference labels $E_{\text{tot}}$ and $\boldsymbol{F}$ from first-principles calculations, the MLFF is generally trained with the energy–force objective:

$$L_0 = \lambda_{\text{E}} L_{\text{E}}(\hat{E}_{\text{tot}}, E_{\text{tot}}) + \lambda_{\text{F}} L_{\text{F}}(\hat{\boldsymbol{F}}, \boldsymbol{F}), \tag{2}$$

where $L_{\text{E}}$ and $L_{\text{F}}$ are typically mean square (or mean absolute) errors weighted by $\lambda_{\text{E}}, \lambda_{\text{F}} \in \mathbb{R}^+$.

**Knowledge Distillation(KD).** In KD, a pretrained teacher model provides auxiliary supervision through an additional loss term $L_{\text{KD}}$. Augmenting the base objective yields the final training objective:

$$L = L_0 + L_{\text{KD}}. \tag{3}$$

**Energy Hessian distillation.** As proposed by Amin et al. (2025), the energy Hessians of the teacher $\boldsymbol{H}^T = \frac{\partial^2 E_{\text{tot}}^T}{\partial \boldsymbol{R}^2}$ can serve as a curvature target. The student matches this curvature by aligning its Hessians:

$$L_{\text{KD}} = \lambda_{\text{H}} L_{\text{H}}(\frac{\partial^2 \hat{E}_{\text{tot}}}{\partial \boldsymbol{R}^2}, \boldsymbol{H}^T), \tag{4}$$

where $\lambda_{\mathrm{H}}$ is a hyperparameter controlling the strength of KD. For models that predict forces directly, $\boldsymbol{H}^T$ can equivalently be realized as the negative Jacobian of the predicted forces $\boldsymbol{H}^T = -\frac{\partial \boldsymbol{F}^T}{\partial \boldsymbol{R}}$. To reduce computational cost, Amin et al. (2025) further supervises only a subsample of Hessian rows through Vector–Jacobian products, which preserves curvature guidance while scaling linearly with the number of sampled rows (see Appendix A.1 for details).

## 3 METHOD

In this section, we first analyze the energy Hessian distillation from a frequency-domain perspective. This objective assigns larger weights to higher-frequency components and smaller weights to lower-frequency components (Section 3.1). Based on this, we combine it with atomic energy supervision, which introduces a frequency-independent spectral floor for the total energy error, thereby directly constraining low-frequency components while retaining strong suppression of high-frequency components (3.2). All proofs are given in the Appendix A.2.

### 3.1 ANALYSIS OF ENERGY HESSIAN DISTILLATION

We analyze how distillation with energy Hessians shapes the error spectrum of energies and forces. Because forces are the negative gradient of energy and Hessians are its second derivative, their errors are intrinsically correlated. We formalize this correlation using a Fourier analysis below.

**Setting and notation.** Let $N$ be the number of atoms and set $d = 3N$ for the number of Cartesian degrees of freedom. Fix a reference configuration $\boldsymbol{R}_\star \in \mathbb{R}^d$ and write displacements $\boldsymbol{x} = \boldsymbol{R} - \boldsymbol{R}_\star \in \mathbb{R}^d$. Let $E_{\mathrm{tot}}^T, E_{\mathrm{tot}}^S : \mathcal{X} \to \mathbb{R}$ denote the total energies of the teacher and the student on a domain $\mathcal{X} \subset \mathbb{R}^d$ containing a neighborhood of $\boldsymbol{0}$. The associated forces and Hessians are defined by:

$$\boldsymbol{F}^T(\boldsymbol{x}) = -\nabla E_{\mathrm{tot}}^T(\boldsymbol{R}_\star + \boldsymbol{x}), \quad \boldsymbol{H}^T(\boldsymbol{x}) = \nabla^2 E_{\mathrm{tot}}^T(\boldsymbol{R}_\star + \boldsymbol{x}), \tag{5}$$

$$\boldsymbol{F}^S(\boldsymbol{x}) = -\nabla E_{\mathrm{tot}}^S(\boldsymbol{R}_\star + \boldsymbol{x}), \quad \boldsymbol{H}^S(\boldsymbol{x}) = \nabla^2 E_{\mathrm{tot}}^S(\boldsymbol{R}_\star + \boldsymbol{x}). \tag{6}$$

We define the energy, force, and Hessian errors by:

$$\delta E_{\mathrm{tot}}(\boldsymbol{x}) = E_{\mathrm{tot}}^S(\boldsymbol{R}_\star + \boldsymbol{x}) - E_{\mathrm{tot}}^T(\boldsymbol{R}_\star + \boldsymbol{x}), \; \delta \boldsymbol{F}(\boldsymbol{x}) = \boldsymbol{F}^S(\boldsymbol{x}) - \boldsymbol{F}^T(\boldsymbol{x}), \; \delta \boldsymbol{H}(\boldsymbol{x}) = \boldsymbol{H}^S(\boldsymbol{x}) - \boldsymbol{H}^T(\boldsymbol{x}). \tag{7}$$

For vectors we use the Euclidean norm $\|\cdot\|_2$, and for matrices the Frobenius norm $\|\cdot\|_F$. Equipping $\mathcal{X}$ with the Lebesgue measure, we define the $L^2$ norms by:

$$\|\delta E_{\mathrm{tot}}\|_{L^2}^2 = \int_{\mathcal{X}} |\delta E_{\mathrm{tot}}(\boldsymbol{x})|^2 d\boldsymbol{x}, \quad \|\delta \boldsymbol{F}\|_{L^2}^2 = \int_{\mathcal{X}} \|\delta \boldsymbol{F}(\boldsymbol{x})\|_2^2 d\boldsymbol{x}, \quad \|\delta \boldsymbol{H}\|_{L^2}^2 = \int_{\mathcal{X}} \|\delta \boldsymbol{H}(\boldsymbol{x})\|_F^2 d\boldsymbol{x}. \tag{8}$$

To simplify the treatment of boundary terms in Fourier expansions, we assume periodic boundary conditions in space. This assumption is natural for crystalline systems. For aperiodic systems, since our interest is typically limited to the rational chemical space within a finite region, it is unnecessary to integrate over the entire space when evaluating errors. In such cases, the boundary can be reasonably approximated as periodic to avoid complications from boundary terms, without affecting the validity of results in the local region.

**Assumptions.** We make the following assumptions used in the analyses.

(A1) (Periodicity) There exists $L > 0$ such that $\mathcal{X}$ is identified with the $d$-dimensional flat torus $\mathbb{T}^d := (\mathbb{R}/L\mathbb{Z})^d$, equipped with the Lebesgue measure. We therefore regard $E_{\mathrm{tot}}^T$ and $E_{\mathrm{tot}}^S$ as L-periodic functions on $\mathbb{T}^d$.

(A2) (Regularity) $E_{\mathrm{tot}}^T, E_{\mathrm{tot}}^S, \delta E_{\mathrm{tot}} \in C^2(\mathbb{T}^d)$. In particular, $\nabla \delta E_{\mathrm{tot}}$ and $\nabla^2 \delta E_{\mathrm{tot}}$ exist pointwise and belong to $L^2(\mathbb{T}^d, d\boldsymbol{x})$.

**Fourier basis and frequencies.** Let $\{\varphi_{\boldsymbol{k}}\}_{\boldsymbol{k} \in \mathbb{Z}^d}$ be the orthonormal Fourier basis of $L^2(\mathbb{T}^d, d\boldsymbol{x})$:

$$\varphi_{\boldsymbol{k}}(\boldsymbol{x}) := L^{-d/2} \exp\left(i\frac{2\pi}{L}\boldsymbol{k}\cdot\boldsymbol{x}\right), \qquad \boldsymbol{k} \in \mathbb{Z}^d, \tag{9}$$

so that $\int_{\mathbb{T}^d} \varphi_{\boldsymbol{k}} \overline{\varphi_{\boldsymbol{\ell}}} \mathrm{d}\boldsymbol{x} = \delta_{\boldsymbol{k}\boldsymbol{\ell}}$. We expand

$$\delta E_{\mathrm{tot}}(\boldsymbol{x}) = \sum_{\boldsymbol{k} \in \mathbb{Z}^d} a_{\boldsymbol{k}} \varphi_{\boldsymbol{k}}(\boldsymbol{x}), \qquad a_{\boldsymbol{k}} = \int_{\mathbb{T}^d} \delta E_{\mathrm{tot}}(\boldsymbol{x}) \overline{\varphi_{\boldsymbol{k}}(\boldsymbol{x})} \mathrm{d}\boldsymbol{x} \in \mathbb{C}, \tag{10}$$

and for real-valued $\delta E_{\mathrm{tot}}$ we have $a_{-\boldsymbol{k}} = \overline{a_{\boldsymbol{k}}}$. Define the angular frequency $\omega_{\boldsymbol{k}} := \frac{2\pi}{L} \|\boldsymbol{k}\|_2$.

We next make explicit how derivative relationships translate into frequency-weighted errors. The following two lemmas formalize this link and set up the spectral identities we will use throughout.

**Lemma 3.1** (Force/Hessian errors are derivatives of the energy error). *For all $\boldsymbol{x} \in \mathbb{T}^d$,*

$$\delta \boldsymbol{F}(\boldsymbol{x}) = -\nabla \delta E_{tot}(\boldsymbol{x}), \qquad \delta \boldsymbol{H}(\boldsymbol{x}) = \nabla^2 \delta E_{tot}(\boldsymbol{x}). \tag{11}$$

This identity lets us study all three errors via a single scalar $\delta E_{\mathrm{tot}}$. With this representation, Parseval–Plancherel turns $L^2$ norms into weighted sums over Fourier coefficients.

**Lemma 3.2** (Parseval–Plancherel identities for $\delta E_{\mathrm{tot}}$, $\delta \boldsymbol{F}$, $\delta \boldsymbol{H}$).

$$\|\delta E_{tot}\|_{L^2}^2 = \sum_{\boldsymbol{k} \in \mathbb{Z}^d} |a_{\boldsymbol{k}}|^2, \tag{12}$$

$$\|\delta \boldsymbol{F}\|_{L^2}^2 = \sum_{\boldsymbol{k} \in \mathbb{Z}^d} \omega_{\boldsymbol{k}}^2 |a_{\boldsymbol{k}}|^2, \tag{13}$$

$$\|\delta \boldsymbol{H}\|_{L^2}^2 = \sum_{\boldsymbol{k} \in \mathbb{Z}^d} \omega_{\boldsymbol{k}}^4 |a_{\boldsymbol{k}}|^2. \tag{14}$$

Hence, energy, force, and Hessian share the same Fourier coefficients $a_{\boldsymbol{k}}$, weighted respectively by $1$, $\omega_{\boldsymbol{k}}^2$, and $\omega_{\boldsymbol{k}}^4$. This frequency weighting underlies all subsequent bounds. The $\omega_{\boldsymbol{k}}^2$ versus $\omega_{\boldsymbol{k}}^4$ weighting immediately yields an $L^2$ control of force error by Hessian error.

**Theorem 3.3** (Force is controlled by Hessian).

$$\|\delta \boldsymbol{F}\|_{L^2}^2 \leq \left(\frac{L}{2\pi}\right)^2 \|\delta \boldsymbol{H}\|_{L^2}^2. \tag{15}$$

*Moreover, if the Fourier expansion of $\delta E$ satisfies a spectral gap $\omega_{\boldsymbol{k}} \geq \Omega_0 > 0$ whenever $a_{\boldsymbol{k}} \neq 0$ and $\boldsymbol{k} \neq \boldsymbol{0}$, then*

$$\|\delta \boldsymbol{F}\|_{L^2}^2 \leq \Omega_0^{-2} \|\delta \boldsymbol{H}\|_{L^2}^2. \tag{16}$$

*The constant $(L/2\pi)$ is optimal and dimension-free, and the equality holds for any single Fourier mode with $\|\boldsymbol{k}\|_2 = 1$.*

This shows that reducing Hessian error necessarily reduces force error, up to a sharp constant. Because higher frequencies are amplified more strongly ($\omega^4$) in the Hessian norm, minimizing Hessian error preferentially suppresses high-frequency components of $\delta E_{\mathrm{tot}}$. The next result quantifies this suppression above any frequency threshold $\Omega$.

**Theorem 3.4** (High-frequency suppression under Hessian-only training). *For any $\Omega > 0$ and define $\mathcal{K}_{\geq \Omega} := \{\boldsymbol{k} \in \mathbb{Z}^d : \omega_{\boldsymbol{k}} \geq \Omega\}$ and $\mathcal{K}_{<\Omega} := \mathbb{Z}^d \setminus \mathcal{K}_{\geq \Omega}$. Then*

$$\sum_{\boldsymbol{k} \in \mathcal{K}_{\geq \Omega}} |a_{\boldsymbol{k}}|^2 \leq \Omega^{-4} \|\delta \boldsymbol{H}\|_{L^2}^2, \tag{17}$$

$$\sum_{\boldsymbol{k} \in \mathcal{K}_{\geq \Omega}} \omega_{\boldsymbol{k}}^2 |a_{\boldsymbol{k}}|^2 \leq \Omega^{-2} \|\delta \boldsymbol{H}\|_{L^2}^2. \tag{18}$$

Thus, Hessian-only training enforces small high-frequency content both in energy error and in the induced force error. However, the same argument provides only weak control at low frequencies, where the $\omega$ weights are small. Decomposing $\delta E_{\mathrm{tot}}$ into low- and high-frequency parts yields the following corollary.

**Corollary 3.5** (Limited control of energy by Hessian-only). *For any $\Omega > 0$,*

$$\|\delta E_{tot}\|_{L^2}^2 = \sum_{\boldsymbol{k} \in \mathcal{K}_{<\Omega}} |a_{\boldsymbol{k}}|^2 + \sum_{\boldsymbol{k} \in \mathcal{K}_{\geq \Omega}} |a_{\boldsymbol{k}}|^2 \leq \sum_{\boldsymbol{k} \in \mathcal{K}_{<\Omega}} |a_{\boldsymbol{k}}|^2 + \Omega^{-4} \|\delta \boldsymbol{H}\|_{L^2}^2. \tag{19}$$

*In particular, the constant mode $a_{\boldsymbol{0}}$ is completely unconstrained by Hessian training.*

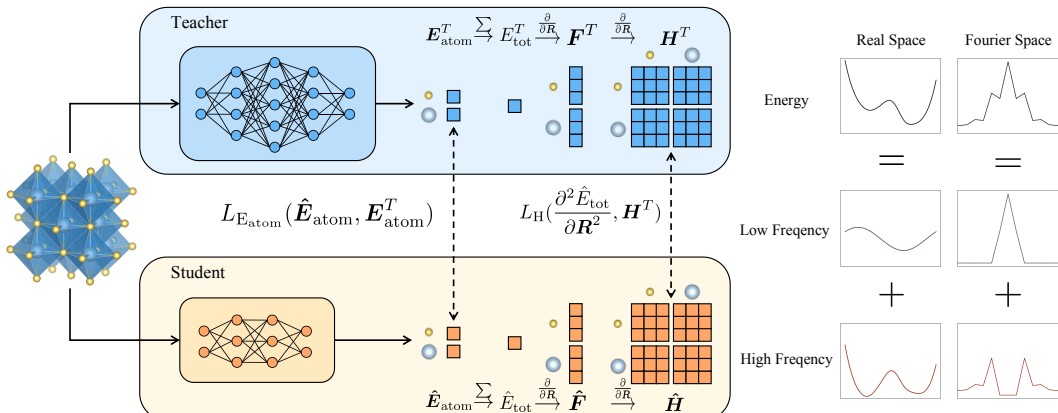

Figure 1: **Joint Atomic Energy–Energy Hessian Distillation.** The knowledge distillation loss includes matching of atomic energies (low-frequency supervision) and energy Hessians (high-frequency supervision) between the teacher and student models.

## 3.2 JOINT ATOMIC ENERGY–ENERGY HESSIAN DISTILLATION

Section 3.1 shows that Hessian-only training strongly damps high-frequency errors but leaves low-frequency components weakly constrained. Here, we augment the objective with atomic energy supervision from the teacher model to improve total energy prediction.

Our distillation objective is as follows:

$$L_{\text{KD}} = \lambda_{\text{E}_{\text{atom}}} L_{\text{E}_{\text{atom}}}(\hat{\boldsymbol{E}}_{\text{atom}}, \boldsymbol{E}_{\text{atom}}^T) + \lambda_{\text{H}} L_{\text{H}}(\frac{\partial^2 \hat{E}_{\text{tot}}}{\partial \boldsymbol{R}^2}, \boldsymbol{H}^T), \tag{20}$$

where $\lambda_{\text{E}_{\text{atom}}}, \lambda_{\text{H}} \in \mathbb{R}^+$ control the relative weights of the two terms. Here, $\boldsymbol{E}_{\text{atom}}^T$ and $\boldsymbol{H}^T$ denote the atomic energies and energy Hessians of the teacher, while $\hat{\boldsymbol{E}}_{\text{atom}}$ and $\hat{E}_{\text{tot}}$ are the student's atomic energies and total energy predictions.

In practice, we adopt the same sampling strategy and implementation settings as in energy Hessian distillation (Amin et al., 2025). The atomic energy term complements curvature matching by providing localized supervision on per-atom contributions. Importantly, supervising atomic energies incurs negligible overhead: it only requires caching the model's intermediate per-atom outputs at inference time, without additional forward passes beyond those already performed for energy/force prediction.

Let atomic energies error be $\delta E_{\text{atom}}^{(i)} = E_{\text{atom}}^{S,(i)} - E_{\text{atom}}^{T,(i)}$ and assume $\delta E_{\text{atom}}^{(i)} \in L^2(\mathbb{T}^d)$. The total energy error is the sum of atomic energies error $\delta E_{\text{tot}} = \sum_{i=1}^N \delta E_{\text{atom}}^{(i)}$. We reanalyze the joint training objectives.

Intuitively, atomic energy matching should impose a uniform, frequency-agnostic penalty, whereas Hessian matching amplifies penalties at higher frequencies. The following theorem makes this decomposition precise by exhibiting an explicit mode-wise lower bound.

**Theorem 3.6** (Atomic energy and energy Hessian supervision induces a uniform spectral floor). *Define the joint objective*

$$L_{\alpha,\beta} := \alpha \sum_{i=1}^N \|\delta E_{atom}^{(i)}\|_{L^2}^2 + \beta \|\nabla^2 \delta E_{\text{tot}}\|_{L^2}^2, \qquad \alpha, \beta > 0. \tag{21}$$

*then*

$$\sum_{\boldsymbol{k} \in \mathbb{Z}^d} (\frac{\alpha}{N} + \beta \omega_{\boldsymbol{k}}^4) |a_{\boldsymbol{k}}|^2 \le L_{\alpha,\beta}. \tag{22}$$

*Hence every frequency component of $\delta E_{\text{tot}}$ is penalized by at least $\alpha/N$, and the constant mode is directly constrained.*

Decomposing the spectrum at an arbitrary threshold $\Omega$ turns the mode-wise bound into controls for low and high frequencies. The corollary below quantifies these two regimes.

**Corollary 3.7** (Uniform low-frequency and high-frequency suppression)**.** *For any $\Omega > 0$ and define* $\mathcal{K}_{\geq \Omega} := \{\boldsymbol{k} \in \mathbb{Z}^d : \omega_{\boldsymbol{k}} \geq \Omega\}$ *and* $\mathcal{K}_{<\Omega} := \mathbb{Z}^d \setminus \mathcal{K}_{\geq \Omega}$. *Then*

$$\sum_{\boldsymbol{k} \in \mathcal{K}_{<\Omega}} |a_{\boldsymbol{k}}|^2 \leq \frac{N}{\alpha} L_{\alpha,\beta}, \tag{23}$$

$$\sum_{\boldsymbol{k} \in \mathcal{K}_{\geq \Omega}} |a_{\boldsymbol{k}}|^2 \leq \frac{1}{\alpha/N + \beta\Omega^4} L_{\alpha,\beta}. \tag{24}$$

*In particular, for the constant mode,* $|a_{\boldsymbol{0}}|^2 \leq (N/\alpha) \, L_{\alpha,\beta}$.

## 4  EXPERIMENTS

To evaluate our proposed method, we present a comprehensive comparison of the three objectives under strictly matched conditions for distilling student models in specific chemical subdomains:

$$\text{No distillation:} \quad L = L_0 = \lambda_\text{E} L_\text{E}(\hat{E}_\text{tot}, E_\text{tot}) + \lambda_\text{F} L_\text{F}(\hat{\boldsymbol{F}}, \boldsymbol{F}),$$

$$\text{Hessian:} \quad L = L_0 + \lambda_\text{H} L_\text{H}\big(\frac{\partial^2 \hat{E}_\text{tot}}{\partial \boldsymbol{R}^2}, \boldsymbol{H}^T\big),$$

$$\text{Atomic Energy + Hessian:} \quad L = L_0 + \lambda_{\text{E}_\text{atom}} L_{\text{E}_\text{atom}}(\hat{\boldsymbol{E}}_\text{atom}, \boldsymbol{E}_\text{atom}^T) + \lambda_\text{H} L_\text{H}\big(\frac{\partial^2 \hat{E}_\text{tot}}{\partial \boldsymbol{R}^2}, \boldsymbol{H}^T\big),$$

where $E_\text{tot}, \boldsymbol{F}$ in $L_0$ are DFT labels from training datasets, while $\boldsymbol{E}_\text{atom}^T, \boldsymbol{H}^T$ are provided by the teacher model.

### 4.1  EXPERIMENTAL SETUP

**Teacher Models.**  We consider three pretrained teachers that span organic and inorganic domains and differ in the architecture and training datasets. MACE-OFF (Kovács et al., 2025) is a short-range, higher-order equivariant message passing potential trained primarily on an augmented subset of SPICE (Eastman et al., 2023; 2024) at the $\omega$B97M–D3(BJ)/def2–TZVPPD level of quantum mechanics, covering neutral organic molecules with elements H, C, N, O, F, P, S, Cl, Br, and I. It provides high-accuracy energies and forces suitable for small molecules and biomolecular fragments. MACE-MP (Batatia et al., 2023) is a universal materials model trained on MPtrj (Deng et al., 2023) of DFT (PBE/GGA+U) relaxation trajectories for $\sim$150,000 inorganic crystals, designed to deliver stable molecular dynamics and transferable accuracy across diverse inorganic systems. Finally, eSEN (Fu et al., 2025) is a recent smooth and expressive equivariant interatomic potential introduced to improve downstream physical-property predictions (e.g., stability, phonons, thermal transport) and trained on MPtrj, sAlex, and OMat24 dataset (Barroso-Luque et al., 2024). For each dataset, we use the teacher trained on that subdomain to generate labels, including atomic energies $\boldsymbol{E}_\text{atom}^T$ and Hessians $\boldsymbol{H}^T$ via second-order derivatives.

**Datasets and Metrics.**  We distill student models on representative subsets from the organic SPICE dataset and the inorganic MPtrj dataset. For SPICE, we use Monomers, Solvated Amino Acids, and Systems with Iodine as three subdomains. For MPtrj, we use $Pm\overline{3}m$ Spacegroup, Systems with Yttrium, and Systems with band gap $\geq$ 5meV. These selections follow previous work (Amin et al., 2025) and span small organic molecules, solvated biomolecular fragments, heavy-atom systems, high-symmetry crystalline configurations, Y-containing materials, and electronically filtered materials, covering both near-equilibrium and perturbed configurations. Primary metrics are: (i) energy MAE (lower is better), reported as total MAE (meV) or per-atom MAE (meV/atom); (ii) force MAE (meV/Å) (lower is better). We also provide a MD stability analysis in Appendix A.9.

**Student Models.**  We adopt three widely used rotational equivariant graph neural networks as students: GemNet-dT, PaiNN, and GemNet-T. GemNet-dT and GemNet-T (Gasteiger et al., 2021) are directional message passing architectures with angle and dihedral angle features designed to capture higher-order geometric correlations in local neighborhoods, while PaiNN (Schütt et al., 2021)

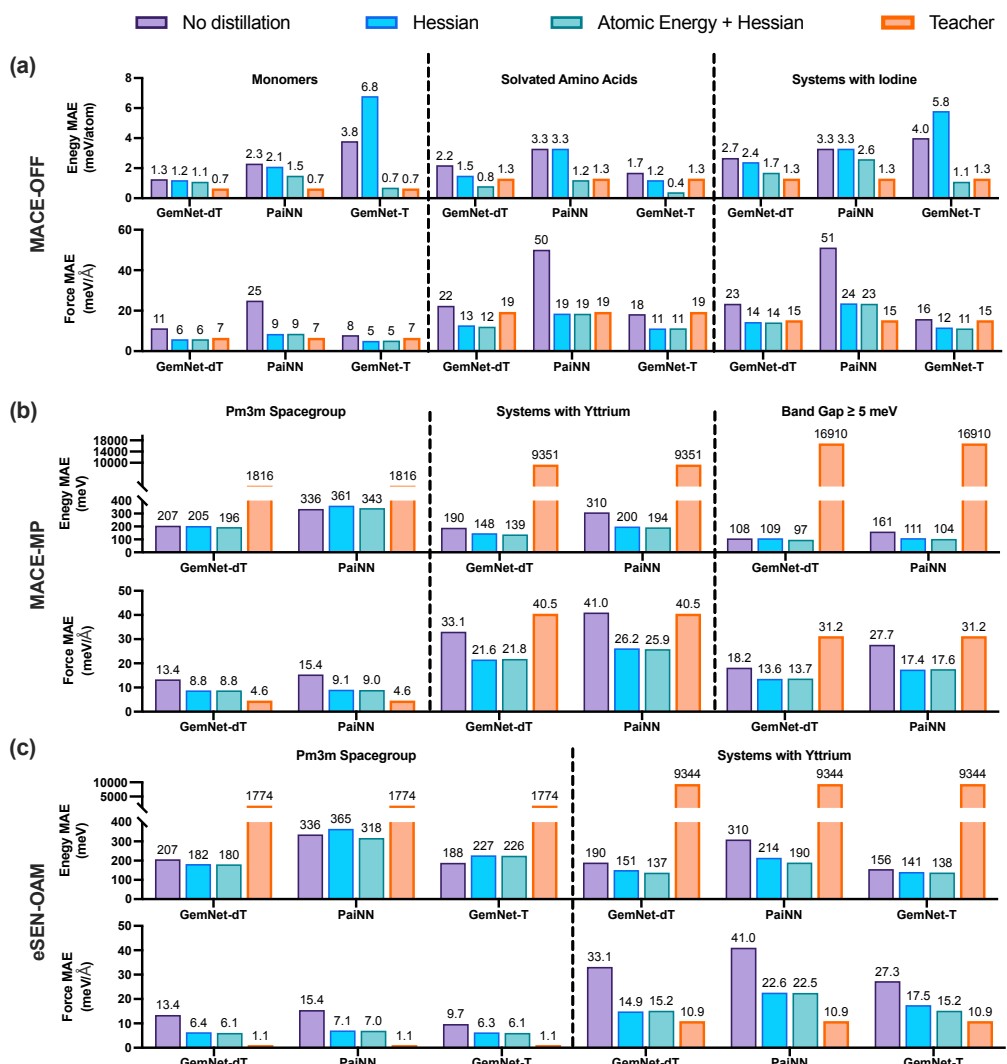

Figure 2: **Accuracy of student models on energy and forces.** (a) Results of distilling MACE-OFF trained on SPICE into student MLFFs. (b) Results of distilling MACE-MP trained on MPtrj into student MLFFs. (c) Results of distilling eSEN-OAM trained on MPtrj into student MLFFs.

is a tensorial message passing network that enforces rotational equivariance through separate scalar and vector channels. Unless otherwise stated, model-specific hyperparameters (e.g., embedding width, number of interaction blocks) are chosen from commonly used configurations validated in prior work. To ensure fairness across three objectives, we use identical training hyperparameters (optimizer, learning rate schedule, batch size, etc.). Training details are listed in Appendix A.3 A.6.

### 4.2 ACCURACY OF STUDENT MODELS

Using MACE-OFF as the teacher, we train each student model separately on each SPICE subset (Monomers, Solvated Amino Acids, Systems with Iodine) and compare their accuracy on energy and forces using the test set. As shown in Figure 2 (a), the joint objective (Atomic Energy + Hessian) consistently improves energy MAE over Hessian distillation by an average of 48.3%, while maintaining comparable force MAE across GemNet-dT, PaiNN, and GemNet-T. Detailed numbers including teacher accuracy are provided in Table 2.

On MPtrj, we use MACE-MP and eSEN-OAM as teachers and train the same students on subsets of MPtrj ($Pm\bar{3}m$ Spacegroup, Systems with Yttrium, and Systems with band gap $\geq$ 5meV). Figure 2

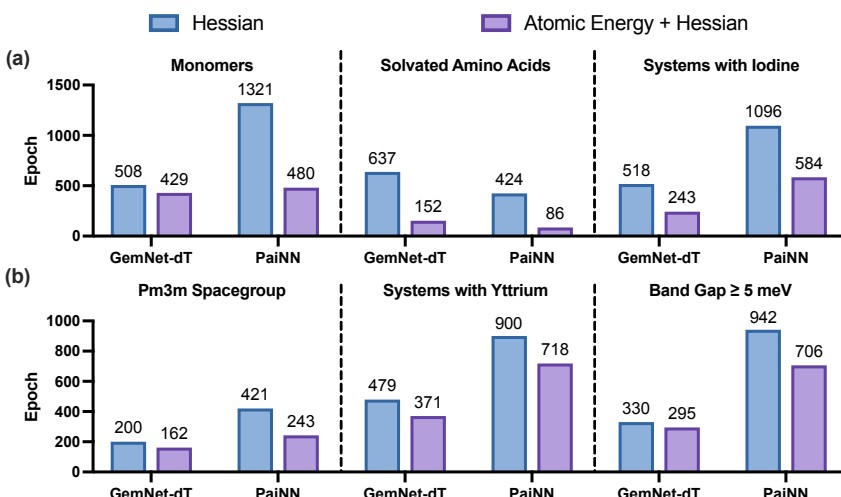

Figure 3: **Epochs to match Hessian final energy MAE.** (a) Results of distilling MACE-OFF. (b) Results of distilling MACE-MP.

(b)(c) shows that the joint objective again outperforms Hessian distillation on energy MAE, with a mean gain of 6.1%, while maintaining comparable force MAE. Detailed numbers including teacher accuracy are provided in Table 3 4. For band gap $\geq$ 5meV subset, we were unable to generate Hessian labels from eSEN-OAM, because only a single sample exhausts the available 80 GB of GPU memory, so results for that configuration are not reported.

## 4.3 Efficiency of Student Models

**Fast Convergency.** We compare the training process between Hessian and Atomic Energy + Hessian. Specifically, we record the training epoch at which each method first reaches the final energy MAE of the Hessian baseline. For this analysis, teachers are MACE-OFF (SPICE) and MACE-MP (MPtrj), and the student models are GemNet-dT and PaiNN. Results are shown in Figure 3. Across datasets and these student architectures, the joint objective consistently attains this energy accuracy in fewer epochs, indicating that adding the atomic energy term accelerates convergence on energy. Meanwhile, the convergence speed for forces is broadly similar between the two objectives throughout the training process.

**Training Overhead.** Compared to Hessian distillation, the extra atomic energy loss requires no additional forward evaluations. It only caches intermediate model outputs, thus incurring minimal overhead. On the Solvated Amino Acids subset of SPICE, using MACE-OFF as the teacher, the additional atomic energy loss increases end-to-end iteration time by less than 0.5%. This negligible overhead, combined with the accuracy gains reported in Figure 2, yields a favorable accuracy-time trade-off. Full experimental details are provided in the Appendix A.5.

**Throughput.** We evaluate the accuracy-throughput trade-off across GemNet-dT student scales on the Solvated Amino Acids subset, varying the number of parameters by adjusting the embedding dimension. Throughput is estimated from single-step molecular dynamics wall-clock time with a 1 fs timestep. Timings are per step and measured after warm-up under an identical hardware platform. Figure 4 shows that, relative to Hessian, the joint Atomic Energy + Hessian objective further improves the balance between energy accuracy and simulation speed across most model sizes. Force accuracy exhibits the same overall trend between the two methods. For the smallest student, the two objectives yield nearly identical energy and force MAE, likely because limited model capacity constrains the benefit obtainable from the joint loss.

## 4.4 Dihedral Scan of the Potential Energy Surface

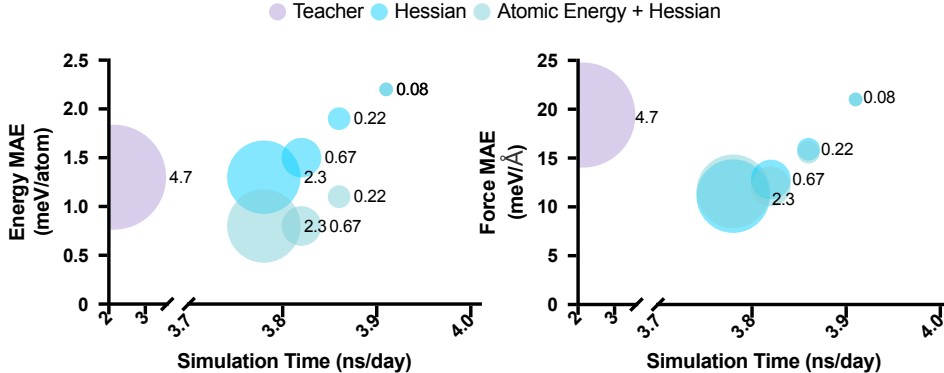

Figure 4: **Accuracy-Throughput trade-off across model scales.** Dot size indicates relative trainable parameters. The label shows parameter count in millions (M). Results are measured on Solvated Amino Acids subset using GemNet-dT student models.

A dihedral scan rotates a molecule around a selected dihedral and evaluates the relative energy at each angle to trace a low-dimensional slice of the potential energy surface (PES), which directly identifies energy minima and thus efficiently finds stable conformations. We conduct a dihedral-scan experiment on the 3-(benzyloxy)pyridin-2-amine (3BPA) (Kovács et al., 2021) dataset to evaluate the extrapolation ability of two distillation strategies on out-of-distribution PES. 3BPA is a drug-like molecule that is large and highly flexible in conformation. The test set comprises "optimized dihedral slices" obtained by systematically rotating key dihedral angles, covering PES regions far from the training distribution. In this experiment, we use GemNet-dT as the student model and, under identical training settings and data splits, perform distillation using Hessian and distillation using both atomic energy and Hessian. We then compute and compare the PES along three representative dihedral slices against DFT references.

The results in Figure 5 show that distillation with both atomic energy and Hessian yields PES curves that align more closely with DFT in the downstream dihedral-scan task. In particular, the Atomic Energy + Hessian model matches DFT better in both the positions and heights of energy barriers, with peak $\gamma$ angles and amplitudes closer to the reference. It also better reproduces the locations and depths of minima, capturing the relative energies and associated angles of low-energy conformations more accurately. These observations indicate that incorporating both atomic energy and Hessian in distillation reconstructs the key geometric features of the PES more faithfully, leading to stronger extrapolation performance in out-of-distribution dihedral-scan scenarios.

## 5 RELATED WORK

**Specilized Machine Learning Force Field.** Specialized MLFFs are data-driven models that approximate a system's potential energy surface and forces from atomic structures. They typically fall into descriptor-based models and GNNs. Descriptor-based models (e.g., DeePMD (Wang et al., 2018; Zeng et al., 2023), NEP (Fan et al., 2022)) construct efficient invariant local descriptors and regress energy and forces. GNNs rely on message passing and geometric inductive biases and can be grouped into invariant feature based models (e.g. SchNet (Schütt et al., 2017), PhysNet (Unke & Meuwly, 2019), DimeNet (Gasteiger et al., 2020), TorchMD-NET (Pelaez et al., 2024)) and equivariant feature based models (e.g. EGNN (Satorras et al., 2021), NequIP (Batzner et al., 2022), MACE (Batatia et al., 2022), Allegro (Musaelian et al., 2023), eSCN (Passaro & Zitnick, 2023), HDGNN (An et al., 2024), GotenNet (Aykent & Xia, 2025),). These architectures trade off computational cost, conservative force vs. direct force parameterization, and locality vs. long-range handling, providing a broad design space for knowledge distillation.

**MLFF Foundation Models.** MLFF foundation models are MLFFs pretrained on diverse, large-scale atomistic datasets to enable broad transfer and stronger zero-shot robustness. The rapid emergence of open datasets has made large-scale pretraining feasible and reproducible. This progress has spurred a wave of general-purpose models, including CHGNet (Deng et al., 2023), MACE-

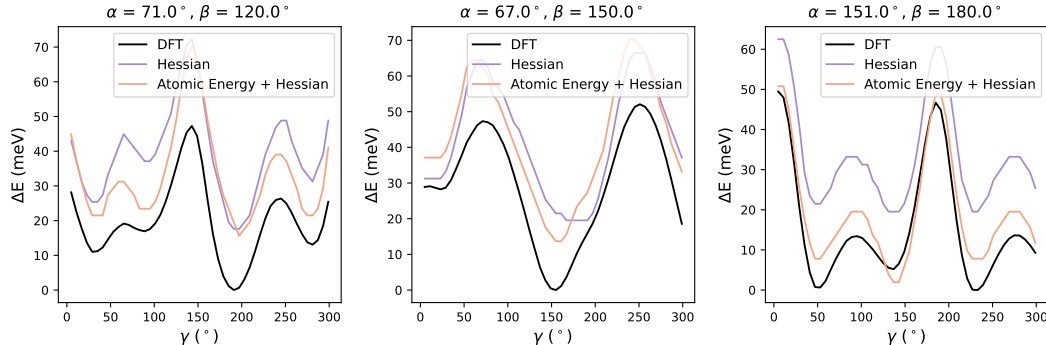

Figure 5: **Dihedral Scan of the Potential Energy Surface (3BPA).** Across three dihedral slices, Atomic Energy + Hessian aligns more closely with DFT, matching peak positions/heights and minima locations/depths better than Hessian.

MP (Batatia et al., 2023), MatterSim (Yang et al., 2024), eqV2 (Liao et al., 2024), SevenNet (Park et al., 2024), eSEN (Fu et al., 2025), ORB (Rhodes et al., 2025), DPA (Zhang et al., 2025), and MACE-OFF (Kovács et al., 2025), which can be used in various tasks including molecular and materials property prediction, structure relaxation, molecular dynamics, and reaction modeling.

**Knowledge Distillation (KD).** Knowledge distillation transfers behaviors from high-capacity teachers to compact students through softened output matching and intermediate-feature alignment (Hinton et al., 2015; Romero et al., 2015; Gou et al., 2021). In natural language processing, distillation compresses large language models through token-, layer-, or sequence-level supervision, exemplified by DistilBERT (Sanh, 2019), TinyBERT (Jiao et al., 2019), and MiniLM (Wang et al., 2020). In computer vision, methods progressed from logit matching to feature, attention, relational, and contrastive objectives (Wang & Yoon, 2021). Related ideas also appear in MLFFs, where FMs serve as teachers, and specialized MLFFs distill intermediate geometric features or physical information from teachers (Ekström Kelvinius et al., 2023; Amin et al., 2025).

## 6 LIMITATIONS AND FUTURE WORK

Our approach inherits limitations from the quality of the teacher: Because atomic energy supervision is derived from the teacher, the student's asymptotic energy accuracy is bounded by the teacher, and gains taper for very small students due to capacity constraints. Moreover, Hessian distillation requires second-order labels, which remain costly to generate and may hinder scalability to larger datasets and higher-capacity teachers. Future work includes calibrating the teacher, exploring teacher ensembles or self-distillation to mitigate teacher bias, and developing cheaper curvature or implicit objectives that approximate Hessian guidance without full second-order labeling.

## 7 CONCLUSION

In this work, we introduce a joint Atomic Energy–Energy Hessian distillation method that augments the state-of-the-art Hessian distillation at minimal cost. Across datasets and teacher-student pairs (MACE-OFF/MACE-MP/eSEN-OAM → GemNet-dT, PaiNN, GemNet-T), the joint loss delivers lower energy MAE with negligible training overhead ($<0.5\%$ in our timing study), maintains comparable force accuracy, accelerates convergence to target energy accuracy, and achieves a more favorable accuracy-throughput trade-off for molecular dynamics across model scales.

REPRODUCIBILITY STATEMENT

To ensure reproducibility, we provide anonymous code repository. All detailed hyperparameters are listed in Appendix A.6, and all datasets used are publicly available. For the theoretical results, we include complete proofs in Appendix A.2.

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

## A APPENDIX

### A.1 HESSIAN DISTILLATION

Amin et al. (2025) introduce Hessian distillation to transfer information from a large MLFF foundation model (teacher) to a smaller, faster specialized student. Beyond the standard energy/force supervision, the objective adds a Hessian alignment term that matches the rows of the teacher's energy Hessians to the negative Jacobian of the student forces with respect to the positions:

$$\mathcal{L}(\phi) = \mathbb{E}_{\mathbf{z}_i, \mathbf{r}_i, \mathbf{H}_i \sim \mathcal{D}_{KD}} \left[ \mathcal{L}_{EF}(\phi) + \lambda_{KD} \|\mathbf{H}_i + \frac{\partial F_\phi(\mathbf{z}_i, \mathbf{r}_i)}{\partial \mathbf{r}}\|_2^2 \right].$$

This method is architecture-agnostic: It applies to teachers and students with different inductive biases, including conservative or direct-force parameterizations and models with or without explicit SO(3) equivariance.

To make Hessian supervision efficient, Amin et al. (2025) supervise only a small, randomly sampled set of Hessian rows per iteration.

$$\mathcal{L}(\phi) = \mathbb{E}_{\mathbf{z}_i, \mathbf{r}_i, \mathbf{H}_i \sim \mathcal{D}_{KD}} \left[ \mathcal{L}_{EF}(\phi) + \lambda_{KD} \cdot \mathbb{E}_{\mathcal{J}_i \sim \mathcal{U}_s(1, 3N)} \left( \frac{1}{s} \sum_{j \in \mathcal{J}_i} \left\| \mathbf{H}_i^{(j)} + \frac{\partial F_\phi^{(j)}(\mathbf{z}_i, \mathbf{r}_i)}{\partial \mathbf{r}} \right\|_2^2 \right) \right].$$

These rows are computed on the student via vector-Jacobian products (VJPs), avoiding the construction of full Hessian matrices. In practice, sampling as few as s=1 row per structure typically preserves accuracy while limiting the training cost to roughly 1.6–2.0× that of undistilled training. On the teacher side, Hessians are precomputed once over the dataset and cached.

### A.2 PROOFS

**Lemma A.1** (Force/Hessian errors are derivatives of the energy error). *For all $\boldsymbol{x} \in \mathbb{T}^d$,*

$$\delta \boldsymbol{F}(\boldsymbol{x}) = -\nabla \delta E_{tot}(\boldsymbol{x}), \qquad \delta \boldsymbol{H}(\boldsymbol{x}) = \nabla^2 \delta E_{tot}(\boldsymbol{x}). \tag{25}$$

*Proof.* By the definitions,

$$\delta \boldsymbol{F}(\boldsymbol{x}) = -\nabla E_{\text{tot}}^S(\boldsymbol{R}_\star + \boldsymbol{x}) + \nabla E_{\text{tot}}^T(\boldsymbol{R}_\star + \boldsymbol{x}) = -\nabla \big[ E_{\text{tot}}^S(\boldsymbol{R}_\star + \boldsymbol{x}) - E_{\text{tot}}^T(\boldsymbol{R}_\star + \boldsymbol{x}) \big], \tag{26}$$

and

$$\delta \boldsymbol{H}(\boldsymbol{x}) = \nabla^2 E_{\text{tot}}^S(\boldsymbol{R}_\star + \boldsymbol{x}) - \nabla^2 E_{\text{tot}}^T(\boldsymbol{R}_\star + \boldsymbol{x}) = \nabla^2 \big[ E_{\text{tot}}^S(\boldsymbol{R}_\star + \boldsymbol{x}) - E_{\text{tot}}^T(\boldsymbol{R}_\star + \boldsymbol{x}) \big]. \tag{27}$$

The interchange of differentiation with subtraction is justified since $E_{\text{tot}}^S, E_{\text{tot}}^T \in C^2(\mathbb{T}^d)$. □

**Lemma A.2** (Parseval–Plancherel identities for $\delta E_{\text{tot}}, \delta \boldsymbol{F}, \delta \boldsymbol{H}$).

$$\|\delta E_{tot}\|_{L^2}^2 = \sum_{\boldsymbol{k} \in \mathbb{Z}^d} |a_{\boldsymbol{k}}|^2, \tag{28}$$

$$\|\delta \boldsymbol{F}\|_{L^2}^2 = \sum_{\boldsymbol{k} \in \mathbb{Z}^d} \omega_{\boldsymbol{k}}^2 |a_{\boldsymbol{k}}|^2, \tag{29}$$

$$\|\delta \boldsymbol{H}\|_{L^2}^2 = \sum_{\boldsymbol{k} \in \mathbb{Z}^d} \omega_{\boldsymbol{k}}^4 |a_{\boldsymbol{k}}|^2. \tag{30}$$

*Proof.* By orthonormality of $\{\varphi_{\boldsymbol{k}}\}$ in $L^2(\mathbb{T}^d, \mathrm{d}\boldsymbol{x})$,

$$\|\delta E_{\text{tot}}\|_{L^2}^2 = \sum_{\boldsymbol{k} \in \mathbb{Z}^d} |a_{\boldsymbol{k}}|^2. \tag{31}$$

For the force, using $\nabla \varphi_{\boldsymbol{k}} = i \frac{2\pi}{L} \boldsymbol{k} \varphi_{\boldsymbol{k}}$,

$$\delta \boldsymbol{F}(\boldsymbol{x}) = -\nabla \delta E_{\text{tot}}(\boldsymbol{x}) = \sum_{\boldsymbol{k}} \big( -i \frac{2\pi}{L} \boldsymbol{k} a_{\boldsymbol{k}} \big) \varphi_{\boldsymbol{k}}(\boldsymbol{x}), \tag{32}$$

hence

$$\|\delta \boldsymbol{F}\|_{L^2}^2 = \sum_{\boldsymbol{k}} \left\| \frac{2\pi}{L} \boldsymbol{k} a_{\boldsymbol{k}} \right\|_2^2 = \sum_{\boldsymbol{k}} \left( \frac{2\pi}{L} \right)^2 \|\boldsymbol{k}\|_2^2 |a_{\boldsymbol{k}}|^2 = \sum_{\boldsymbol{k}} \omega_{\boldsymbol{k}}^2 |a_{\boldsymbol{k}}|^2. \tag{33}$$

For the Hessian, since $\partial_{ij}^2 \varphi_{\boldsymbol{k}} = -\left( \frac{2\pi}{L} \right)^2 k_i k_j \varphi_{\boldsymbol{k}}$, we get

$$\delta \boldsymbol{H}(\boldsymbol{x}) = \nabla^2 \delta E_{\text{tot}}(\boldsymbol{x}) = \sum_{\boldsymbol{k}} \left( - \left( \frac{2\pi}{L} \right)^2 \boldsymbol{k} \boldsymbol{k}^\top a_{\boldsymbol{k}} \right) \varphi_{\boldsymbol{k}}(\boldsymbol{x}), \tag{34}$$

and thus

$$\|\delta \boldsymbol{H}\|_{L^2}^2 = \sum_{\boldsymbol{k}} \left\| \left( \frac{2\pi}{L} \right)^2 \boldsymbol{k} \boldsymbol{k}^\top a_{\boldsymbol{k}} \right\|_F^2 = \sum_{\boldsymbol{k}} \left( \frac{2\pi}{L} \right)^4 \sum_{i,j} k_i^2 k_j^2 |a_{\boldsymbol{k}}|^2 = \sum_{\boldsymbol{k}} \omega_{\boldsymbol{k}}^4 |a_{\boldsymbol{k}}|^2. \tag{35}$$

Here we use $\sum_{i,j} k_i^2 k_j^2 = (\sum_i k_i^2)^2 = \|\boldsymbol{k}\|_2^4$. $\qquad\square$

**Theorem A.3** (Force is controlled by Hessian).

$$\|\delta \boldsymbol{F}\|_{L^2}^2 \le \left( \frac{L}{2\pi} \right)^2 \|\delta \boldsymbol{H}\|_{L^2}^2. \tag{36}$$

*Moreover, if the Fourier expansion of $\delta E$ satisfies a spectral gap $\omega_{\boldsymbol{k}} \ge \Omega_0 > 0$ whenever $a_{\boldsymbol{k}} \neq 0$ and $\boldsymbol{k} \neq \boldsymbol{0}$, then*

$$\|\delta \boldsymbol{F}\|_{L^2}^2 \le \Omega_0^{-2} \|\delta \boldsymbol{H}\|_{L^2}^2. \tag{37}$$

*The constant $(L/2\pi)$ is optimal and dimension-free, and the equality holds for any single Fourier mode with $\|\boldsymbol{k}\|_2 = 1$.*

*Proof.* By Parseval–Plancherel identities,

$$\|\delta \boldsymbol{F}\|_{L^2}^2 = \sum_{\boldsymbol{k} \in \mathbb{Z}^d} \omega_{\boldsymbol{k}}^2 |a_{\boldsymbol{k}}|^2 = \sum_{\boldsymbol{k} \neq \boldsymbol{0}} \omega_{\boldsymbol{k}}^2 |a_{\boldsymbol{k}}|^2, \qquad \|\delta \boldsymbol{H}\|_{L^2}^2 = \sum_{\boldsymbol{k} \in \mathbb{Z}^d} \omega_{\boldsymbol{k}}^4 |a_{\boldsymbol{k}}|^2 = \sum_{\boldsymbol{k} \neq \boldsymbol{0}} \omega_{\boldsymbol{k}}^4 |a_{\boldsymbol{k}}|^2. \tag{38}$$

Since $\omega_{\boldsymbol{k}} \ge \frac{2\pi}{L}$ for $\boldsymbol{k} \neq \boldsymbol{0}$, $\omega_{\boldsymbol{k}}^2 \le \left( \frac{L}{2\pi} \right)^2 \omega_{\boldsymbol{k}}^4$, summing gives equation 36. If $\omega_{\boldsymbol{k}} \ge \Omega_0$ on the Fourier expansion, then $\omega_{\boldsymbol{k}}^2 \le \Omega_0^{-2} \omega_{\boldsymbol{k}}^4$, yielding equation 37. Optimality follows by taking a single mode with $\|\boldsymbol{k}\|_2 = 1$. $\qquad\square$

**Theorem A.4** (High-frequency suppression under Hessian-only training). *For any $\Omega > 0$ and define $\mathcal{K}_{\ge \Omega} := \{\boldsymbol{k} \in \mathbb{Z}^d : \omega_{\boldsymbol{k}} \ge \Omega\}$ and $\mathcal{K}_{<\Omega} := \mathbb{Z}^d \setminus \mathcal{K}_{\ge \Omega}$. Then*

$$\sum_{\boldsymbol{k} \in \mathcal{K}_{\ge \Omega}} |a_{\boldsymbol{k}}|^2 \le \Omega^{-4} \|\delta \boldsymbol{H}\|_{L^2}^2, \tag{39}$$

$$\sum_{\boldsymbol{k} \in \mathcal{K}_{\ge \Omega}} \omega_{\boldsymbol{k}}^2 |a_{\boldsymbol{k}}|^2 \le \Omega^{-2} \|\delta \boldsymbol{H}\|_{L^2}^2. \tag{40}$$

*Proof.* Using $\|\delta \boldsymbol{H}\|_{L^2}^2 = \sum_{\boldsymbol{k}} \omega_{\boldsymbol{k}}^4 |a_{\boldsymbol{k}}|^2$,

$$\sum_{\boldsymbol{k} \in \mathcal{K}_{\ge \Omega}} |a_{\boldsymbol{k}}|^2 \le \Omega^{-4} \sum_{\boldsymbol{k} \in \mathcal{K}_{\ge \Omega}} \omega_{\boldsymbol{k}}^4 |a_{\boldsymbol{k}}|^2 \le \Omega^{-4} \|\delta \boldsymbol{H}\|_{L^2}^2, \tag{41}$$

giving equation 39. Similarly,

$$\sum_{\boldsymbol{k} \in \mathcal{K}_{\ge \Omega}} \omega_{\boldsymbol{k}}^2 |a_{\boldsymbol{k}}|^2 = \sum_{\boldsymbol{k} \in \mathcal{K}_{\ge \Omega}} \omega_{\boldsymbol{k}}^{-2} \omega_{\boldsymbol{k}}^4 |a_{\boldsymbol{k}}|^2 \le \Omega^{-2} \sum_{\boldsymbol{k} \in \mathcal{K}_{\ge \Omega}} \omega_{\boldsymbol{k}}^4 |a_{\boldsymbol{k}}|^2 \le \Omega^{-2} \|\delta \boldsymbol{H}\|_{L^2}^2, \tag{42}$$

which is equation 40. $\qquad\square$

**Corollary A.5** (Limited control of energy by Hessian-only). *For any $\Omega > 0$,*

$$\|\delta E_{tot}\|_{L^2}^2 = \sum_{\boldsymbol{k} \in \mathcal{K}_{<\Omega}} |a_{\boldsymbol{k}}|^2 + \sum_{\boldsymbol{k} \in \mathcal{K}_{\ge \Omega}} |a_{\boldsymbol{k}}|^2 \le \sum_{\boldsymbol{k} \in \mathcal{K}_{<\Omega}} |a_{\boldsymbol{k}}|^2 + \Omega^{-4} \|\delta \boldsymbol{H}\|_{L^2}^2. \tag{43}$$

*In particular, the constant mode $a_{\boldsymbol{0}}$ is completely unconstrained by Hessian training.*

**Theorem A.6** (Atomic energy and energy Hessian supervision induces a uniform spectral floor).
*Define the joint objective*

$$L_{\alpha,\beta} := \alpha \sum_{i=1}^{N} \|\delta E_{atom}^{(i)}\|_{L^2}^2 + \beta \|\nabla^2 \delta E_{\text{tot}}\|_{L^2}^2, \qquad \alpha, \beta > 0. \tag{44}$$

*then*

$$\sum_{\boldsymbol{k} \in \mathbb{Z}^d} (\frac{\alpha}{N} + \beta \omega_{\boldsymbol{k}}^4)|a_{\boldsymbol{k}}|^2 \leq L_{\alpha,\beta}. \tag{45}$$

*Hence every frequency component of $\delta E_{\text{tot}}$ is penalized by at least $\alpha/N$, and the constant mode is directly constrained.*

*Proof.* By Cauchy–Schwarz,

$$\sum_{i=1}^{N} \|\delta E_{\text{atom}}^{(i)}\|_{L^2}^2 \geq \frac{1}{N} \| \sum_{i=1}^{N} \delta E_{\text{atom}}^{(i)}\|_{L^2}^2 = \frac{1}{N} \sum_{\boldsymbol{k}} |a_{\boldsymbol{k}}|^2. \tag{46}$$

By Parseval–Plancherel,

$$\|\nabla^2 \delta E_{\text{tot}}\|_{L^2}^2 = \sum_{\boldsymbol{k}} \omega_{\boldsymbol{k}}^4 |a_{\boldsymbol{k}}|^2. \tag{47}$$

Summing the two contributions yields equation 45. $\qquad\square$

**Corollary A.7** (Uniform low-frequency and high-frequency suppression). *For any $\Omega > 0$ and define $\mathcal{K}_{\geq\Omega} := \{\boldsymbol{k} \in \mathbb{Z}^d : \omega_{\boldsymbol{k}} \geq \Omega\}$ and $\mathcal{K}_{<\Omega} := \mathbb{Z}^d \setminus \mathcal{K}_{\geq\Omega}$. Then*

$$\sum_{\boldsymbol{k} \in \mathcal{K}_{<\Omega}} |a_{\boldsymbol{k}}|^2 \leq \frac{N}{\alpha} L_{\alpha,\beta}, \tag{48}$$

$$\sum_{\boldsymbol{k} \in \mathcal{K}_{\geq\Omega}} |a_{\boldsymbol{k}}|^2 \leq \frac{1}{\alpha/N + \beta\Omega^4} L_{\alpha,\beta}. \tag{49}$$

*In particular, for the constant mode, $|a_{\boldsymbol{0}}|^2 \leq (N/\alpha) L_{\alpha,\beta}$.*

### A.3 ABLATION ON ATOMIC ENERGY DISTILLATION WEIGHT

We perform a grid scan over $\lambda_{E_{\text{atom}}} \in \{0, 1, 10, 20, 100, 1000\}$ with all other training settings fixed to isolate its effect. On the Solvated Amino Acids subset of SPICE, using GemNet-dT as the student, the results (see table 1) show that $\lambda_{E_{\text{atom}}} = 20$ achieves the best balance between energy accuracy and force accuracy.

Table 1: Energy MAE (meV/atom) and force MAE (meV/Å) achieved by Atomic Energy + Hessian distillation on the Solvated Amino Acid subset of SPICE, using different values of $\lambda_{E_{\text{atom}}}$.

| $\lambda_{E_{\text{atom}}}$ | **Energy MAE** | **Force MAE** |
|---|---|---|
| 0 | 1.5 | 12.8 |
| 1 | 1.1 | 12.4 |
| 10 | 0.9 | 12.2 |
| 20 | **0.8** | **12.1** |
| 100 | 0.9 | 12.4 |
| 1000 | 1.0 | 14.4 |

We perform a similar sweep for MPTrj and find that $\lambda_{E_{\text{atom}}} = 10$ is optimal.

## A.4 ACCURACY RESULTS

Using MACE-OFF as the teacher, we train each student model separately on each SPICE subset (Monomers, Solvated Amino Acids, Systems with Iodine). Each dataset is split into train, validation, and test sets. All metrics are reported on the test set. "No distillation" baselines are taken from previous work (Amin et al., 2025). For the conservative student GemNet-T, we use the Hessian results reported by Amin et al. (2025). For the non-conservative students GemNet-dT and PaiNN, the Hessian results in Amin et al. (2025) provide additional gradient supervision on the energy head. We re-evaluate pure Hessian distillation under the same experimental setting for fair comparison. The accuracy across students and datasets is summarized in Table 2. To isolate the impact of adding atomic energy supervision to Hessians, we report the relative change $\Delta\% = 100 \times (\text{MAE}_{\text{Atomic Energy + Hessian}} - \text{MAE}_{\text{Hessian}})/\text{MAE}_{\text{Hessian}}$ for both energy and forces. Our method consistently improves energy accuracy over both the No distillation and Hessian baselines while maintaining force accuracy comparable to Hessian distillation. In most cases, our distilled models also outperform teacher models, likely because they can allocate their full capacity to a targeted subdomain of the chemical space, as noted by Amin et al. (2025).

Table 2: **Results of distilling MACE-OFF trained on SPICE into student MLFFs.** (T) indicates teacher model, while (S) indicates student model. The percentages in parentheses for the Atomic Energy + Hessian results indicate the change relative to the Hessian.

| Subset | Size | Model (Parameter Count) | Method | Energy MAE ($\downarrow$) (meV/atom) | Force MAE ($\downarrow$) (meV/Å) |
|---|---|---|---|---|---|
| Monomers | 14,331 | (T) MACE-OFF Large (4.7M) | Pretrained | 0.65 | 6.6 |
| | | (S) GemNet-dT (0.67M) | No distillation | 1.27 | 11.3 |
| | | | Hessian | 1.2 | **5.9** |
| | | | Atomic Energy + Hessian (ours) | **1.1** (-8.3%) | **5.9** (0.0%) |
| | | (S) PaiNN (1.0M) | No distillation | 2.3 | 25.0 |
| | | | Hessian | 2.1 | **8.5** |
| | | | Atomic Energy + Hessian (ours) | **1.5** (-28.6%) | 8.6 (+1.2%) |
| | | (S) GemNet-T (0.57M) | No distillation | 3.8 | 7.9 |
| | | | Hessian | 6.8 | **5.1** |
| | | | Atomic Energy + Hessian (ours) | **0.7** (-89.7%) | 5.2 (+2.0%) |
| Solvated Amino Acids | 805 | (T) MACE-OFF Large (4.7M) | Pretrained | 1.3 | 19.4 |
| | | (S) GemNet-dT (0.67M) | No distillation | 2.2 | 22.4 |
| | | | Hessian | 1.5 | 12.8 |
| | | | Atomic Energy + Hessian (ours) | **0.8** (-46.7%) | **12.1** (-5.5%) |
| | | (S) PaiNN (1.0M) | No distillation | 3.3 | 50.1 |
| | | | Hessian | 3.3 | 18.6 |
| | | | Atomic Energy + Hessian (ours) | **1.2** (-63.6%) | **18.5** (-0.5%) |
| | | (S) GemNet-T (0.57M) | No distillation | 1.7 | 18.3 |
| | | | Hessian | 1.2 | **11.2** |
| | | | Atomic Energy + Hessian (ours) | **0.4** (-66.7%) | 11.3 (+0.9%) |
| Systems with Iodine | 11,171 | (T) MACE-OFF Large (4.7M) | Pretrained | 1.3 | 15.3 |
| | | (S) GemNet-dT (0.67M) | No distillation | 2.68 | 23.4 |
| | | | Hessian | 2.4 | 14.4 |
| | | | Atomic Energy + Hessian (ours) | **1.7** (-29.2%) | **14.2** (-1.4%) |
| | | (S) PaiNN (1.0M) | No distillation | 3.3 | 51.2 |
| | | | Hessian | 3.3 | 23.6 |
| | | | Atomic Energy + Hessian (ours) | **2.6** (-21.2%) | **23.4** (-0.8%) |
| | | (S) GemNet-T (0.57M) | No distillation | 4.0 | 15.9 |
| | | | Hessian | 5.8 | 11.7 |
| | | | Atomic Energy + Hessian (ours) | **1.1** (-81.0%) | **11.2** (-4.3%) |

On MPtrj, we use MACE-MP and eSEN-OAM as teachers and train the same students on subsets of MPtrj ($Pm\bar{3}m$ Spacegroup, Systems with Yttrium, and Systems with band gap $\geq$ 5meV). Due to the reasons mentioned above, the Hessian baseline of non-conservative students GemNet-dT and PaiNN is re-measured without using gradient supervision on the energy head. The accuracy across students and datasets is summarized in Table 3 4. Notably, the students already surpass the teachers on energy accuracy by a wide margin, which may explain why the atomic energy term yields a smaller gain than on SPICE. Nevertheless, the joint loss still further improves energy accuracy.

Table 3: **Results of distilling MACE-MP trained on MPtrj into student MLFFs.** (T) indicates teacher model, while (S) indicates student model. The percentages in parentheses for the Atomic Energy + Hessian results indicate the change relative to the Hessian.

| Subset | Size | Model (Parameter Count) | Method | Energy MAE (↓) (meV) | Force MAE (↓) (meV/Å) |
|---|---|---|---|---|---|
| $Pm\bar{3}m$ Spacegroup | 9,725 | (T) MACE-MP0 (15.8 M) | Pretrained | 1815.5 | 4.6 |
| | | (S) GemNet-dT (0.67M) | No distillation | 206.5 | 13.4 |
| | | | Hessian | 204.7 | **8.8** |
| | | | Atomic Energy + Hessian (ours) | **195.9** (-4.3%) | **8.8** (0.0%) |
| | | (S) PaiNN (1.0M) | No distillation | **335.9** | 15.4 |
| | | | Hessian | 361.1 | 9.1 |
| | | | Atomic Energy + Hessian (ours) | 342.8 (-5.1%) | **9.0** (-1.1%) |
| Systems with Yttrium | 30,436 | (T) MACE-MP0 (15.8 M) | Pretrained | 9351.1 | 40.5 |
| | | (S) GemNet-dT (0.67M) | No distillation | 190.0 | 33.1 |
| | | | Hessian | 148.1 | **21.6** |
| | | | Atomic Energy + Hessian (ours) | **138.5** (-6.5%) | 21.8 (+0.9%) |
| | | (S) PaiNN (1.0M) | No distillation | 309.6 | 41.0 |
| | | | Hessian | 200.4 | 26.2 |
| | | | Atomic Energy + Hessian (ours) | **193.8** (-3.3%) | **25.9** (-1.1%) |
| Band Gap ≥ 5 meV | 36,150 | (T) MACE-MP0 (15.8 M) | Pretrained | 16909.8 | 31.2 |
| | | (S) GemNet-dT (0.67M) | No distillation | 108.4 | 18.2 |
| | | | Hessian | 109.1 | **13.6** |
| | | | Atomic Energy + Hessian (ours) | **97.4** (-10.7%) | 13.7 (+0.7%) |
| | | (S) PaiNN (1.0M) | No distillation | 161.1 | 27.7 |
| | | | Hessian | 110.8 | **17.4** |
| | | | Atomic Energy + Hessian (ours) | **103.8** (-6.3%) | 17.6 (+1.1%) |

Table 4: **Results of distilling eSEN-OAM trained on MPtrj into student MLFFs.** (T) indicates teacher model, while (S) indicates student model. The percentages in parentheses for the Atomic Energy + Hessian results indicate the change relative to the Hessian.

| Subset | Size | Model (Parameter Count) | Method | Energy MAE (↓) (meV) | Force MAE (↓) (meV/Å) |
|---|---|---|---|---|---|
| $Pm\bar{3}m$ Spacegroup | 9,725 | (T) eSEN-30M-OAM (30.2 M) | Pretrained | 1774.3 | 1.1 |
| | | (S) GemNet-dT (0.67M) | No distillation | 206.5 | 13.4 |
| | | | Hessian | 181.6 | 6.4 |
| | | | Atomic Energy + Hessian (ours) | **180.3** (-0.7%) | **6.1** (-4.7%) |
| | | (S) PaiNN (1.0M) | No distillation | 335.9 | 15.4 |
| | | | Hessian | 365.2 | 7.1 |
| | | | Atomic Energy + Hessian (ours) | **317.6** (-13.0%) | **7.0** (-1.4%) |
| | | (S) GemNet-T (0.57M) | No distillation | **188.3** | 9.7 |
| | | | Hessian | 227.2 | 6.3 |
| | | | Atomic Energy + Hessian (ours) | 225.5 (-0.7%) | **6.1** (-3.2%) |
| Systems with Yttrium | 30,436 | (T) eSEN-30M-OAM (30.2 M) | Pretrained | 9344.1 | 10.9 |
| | | (S) GemNet-dT (0.67M) | No distillation | 190.0 | 33.1 |
| | | | Hessian | 150.5 | **14.9** |
| | | | Atomic Energy + Hessian (ours) | **137.2** (-8.8%) | 15.2 (+2.0%) |
| | | (S) PaiNN (1.0M) | No distillation | 309.6 | 41.0 |
| | | | Hessian | 214.4 | 22.6 |
| | | | Atomic Energy + Hessian (ours) | **189.8** (-11.5%) | **22.5** (-0.4%) |
| | | (S) GemNet-T (0.57M) | No distillation | 155.7 | 27.3 |
| | | | Hessian | 140.5 | 17.5 |
| | | | Atomic Energy + Hessian (ours) | **137.8** (-1.9%) | **15.2** (-13.1%) |

## A.5 TRAINING OVERHEAD

Taking the Solvated Amino Acids subset of SPICE as a case study, we measure the training overhead of adding the atomic energy term. Figure 6 shows that the extra loss increases end-to-end iteration time by under 0.5%. Timings cover data loading, neighbor list construction, forward/backward passes, and optimizer steps, and were recorded post warm-up under identical hardware platform and

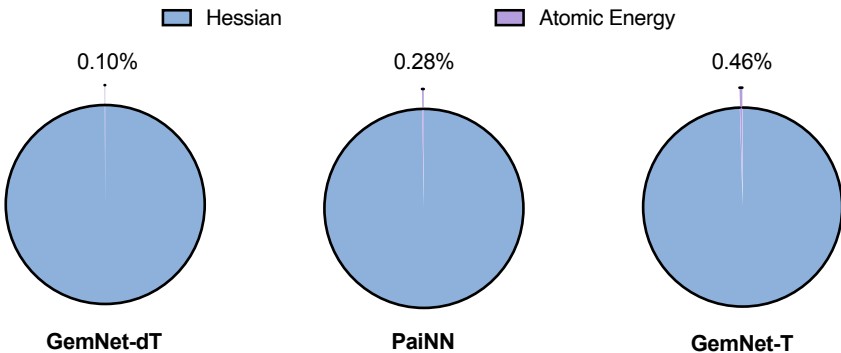

Figure 6: Training overhead from atomic energy supervision.

batch settings. Coupled with the accuracy gains in Figure 2, this yields a favorable accuracy-time trade-off.

### A.6 TRAINING DETAILS

We provide training details for GemNet-dT, GemNet-T, PaiNN models used in this work. Unless otherwise noted, The same hyperparameters are used for No distillation, Hessian and Atomic Energy + Hessian training to ensure fair comparisons.

Table 5 lists the architectural settings for GemNet-dT and GemNet-T, including the basis sizes, embedding dimensions, number of blocks, cutoff, neighbor cap, and activation/initialization choices aligned with the prior stable configurations.

Table 5: Hyperparameters for GemNet-dT and GemNet-T student models.

| Parameter | Value |
|---|---|
| Number of Spherical | 7 |
| Radial Basis Functions | 6 |
| Blocks | 4 |
| Atom Embedding Size | 64 |
| Edge Embedding Size | 64 |
| Triplet Embedding Size | 32 |
| RBF Embedding Size | 16 |
| CBF Embedding Size | 16 |
| Bilinear Triplet Embedding Size | 64 |
| Number Before Skip | 1 |
| Number After Skip | 1 |
| Number of Concatenations | 1 |
| Number of Atoms | 2 |
| Cutoff | 5.0 (SPICE) / 6.0 (MPtrj) |
| Maximum Neighbors | 50 |
| RBF Function | Gaussian |
| Envelope Function | Polynomial (Exponent: 5) |
| CBF Function | Spherical Harmonics |
| Output Initialization | HeOrthogonal |
| Activation Function | SiLU |

Table 6 are the architecture hyperparameters for PaiNN student models. For the MPtrj dataset, the three slash-separated cutoffs are used for $Pm\bar{3}m$ Spacegroup, Systems with Yttrium, and Systems with band gap $\geq 5$meV, respectively.

Table 6: Hyperparameters for PaiNN student models.

| Parameter | Value |
|---|---|
| Hidden Channels | 128 |
| Layers | 4 |
| Radial Basis Functions | 128 |
| Cutoff | 12.0 (SPICE) / [14.0 / 16.0 / 6.0] (MPtrj) |
| Maximum Neighbors | 50 |

Tables 7 8 9 10 summarize optimization schedules, total epochs, loss weights, and batch sizes. We use AdamW with AMSGrad, ReduceLROnPlateau scheduling, gradient clipping, and EMA. Training epoch are subset- and model-specific to match dataset size.

Table 7: Optimization hyperparameters for student models.

| Parameter | GemNet-dT/GemNet-T | PaiNN |
|---|---|---|
| Initial Learning Rate | 0.001 | 0.001 |
| Optimizer | AdamW | AdamW |
| Weight Decay | 0.000002 | 0.000002 |
| Amsgrad | True | True |
| Adam epsilon | 1.e-7 | 1.e-7 |
| Scheduler | ReduceLROnPlateau | ReduceLROnPlateau |
| Patience | 5 | 10 |
| Factor | 0.8 | 0.8 |
| Minimum Learning Rate | 0.000001 | 0.000005 |
| EMA Decay | 0.999 | 0.999 |
| Clip Gradient Norm | 10 | 10 |

Table 8: Training epochs for student models.

| Subset | Model | Epochs |
|---|---|---|
| Monomers | GemNet-dT | 600 |
| | PaiNN | 1400 |
| | GemNet-T | 500 |
| Solvated Amino Acids | GemNet-dT | 2000 |
| | PaiNN | 1000 |
| | GemNet-T | 2500 |
| Systems with Iodine | GemNet-dT | 600 |
| | PaiNN | 1500 |
| | GemNet-T | 600 |
| $Pm\bar{3}m$ Spacegroup | GemNet-dT | 500 |
| | PaiNN | 500 |
| | GemNet-T | 500 |
| Systems with Yttrium | GemNet-dT | 500 |
| | PaiNN | 1000 |
| | GemNet-T | 500 |
| Band Gap $\geq$ 5 meV | GemNet-dT | 350 |
| | PaiNN | 1000 |

Table 9: Loss weights for student models.

| Subset | $\lambda_E$ | $\lambda_F$ | $\lambda_{E_{\text{atom}}}$ | $\lambda_H$ |
|---|---|---|---|---|
| Monomers | 5 | 100 | 20 | 400 |
| Solvated Amino Acids | 5 | 100 | 20 | 400 |
| Structures with Iodine | 5 | 100 | 20 | 400 |
| $Pm\overline{3}m$ Spacegroup | 5 | 100 | 10 | 200 |
| Structures with Yttrium | 5 | 100 | 10 | 200 |
| Band gap $\geq$ 5meV | 5 | 100 | 10 | 200 |

Table 10: Training batch size for student models.

| Subset | GemNet-dT/GemNet-T | PaiNN |
|---|---|---|
| Monomers | 4 | 8 |
| Solvated Amino Acids | 4 | 4 |
| Structures with Iodine | 4 | 8 |
| $Pm\overline{3}m$ Spacegroup | 16 | 16 |
| Structures with Yttrium | 16 | 16 |
| Band Gap $\geq$ 5meV | 32 | 32 |

Table 11 shows the number of rows sampled from Hessian used in each training iteration. These settings follow previous work (Amin et al., 2025).

Table 11: Number of rows sampled from Hessian.

| Subset | GemNet-dT/GemNet-T | PaiNN |
|---|---|---|
| Monomers | 4 | 4 |
| Solvated Amino Acids | 1 | 1 |
| Structures with Iodine | 4 | 4 |
| $Pm\overline{3}m$ Spacegroup | 4 | 4 |
| Structures with Yttrium | 1 | 4 |
| Band Gap $\geq$ 5meV | 1 | 4 |

## A.7 ABLATION ON DFT LABELS

We conduct an ablation study on the Solvated Amino Acids dataset to evaluate the role of explicit DFT labels. The student model is GemNet-dT and all training hyperparameters, data splits, and evaluation protocols match those in the main experiments. The experimental results in the Table 12 show that removing the labels from the DFT reduces accuracy. Furthermore, since the total energy $E$ is the sum of atomic energies $E_{\text{atom}}$, we also experiment with removing only the DFT energy labels. The results likewise showed decreased accuracy, which may be due to inaccuracies in the teacher model's energy predictions, suggesting that supervision from the DFT energy is still necessary. Another possible approach is to replace the DFT force labels with the forces predicted by the teacher model. However, previous studies Amin et al. (2025) have demonstrated that this substitution also results in degraded performance. In summary, partially or completely removing DFT supervision, or replacing it with the teacher model's predictions, all lead to lower accuracy. Therefore, we adopt a strategy that combines both DFT energy and force supervision together with the distillation loss.

Table 12: Energy MAE (meV/atom) and force MAE (meV/Å) on the Solvated Amino Acid subset of SPICE, comparing training with and without DFT supervision.

| $\lambda_E$ | $\lambda_F$ | $\lambda_{E_{\text{atom}}}$ | $\lambda_H$ | **Energy MAE** | **Force MAE** |
|---|---|---|---|---|---|
| 5 | 100 | 0 | 0 | 2.2 | 22.4 |
| 0 | 0 | 20 | 400 | 1.4 | 46.4 |
| 0 | 100 | 20 | 400 | 1.1 | **12.1** |
| 5 | 100 | 20 | 400 | **0.8** | 12.1 |

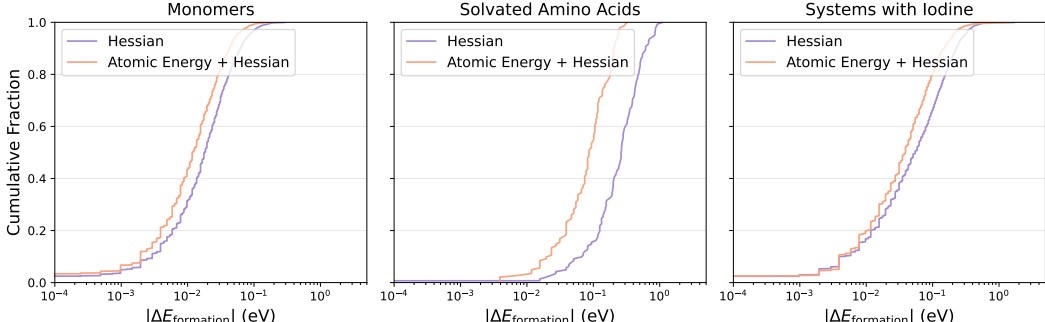

Figure 7: **Cumulative distribution of formation energy absolute errors.** Results are shown for Monomers, Solvated Amino Acids, and Systems with Iodine.

### A.8 FORMATION ENERGY

Stability is a prerequisite for using any material in applications. The formation energy is the key quantity for assessing relative thermodynamic stability and the tendency toward decomposition. It measures the energy difference between a compound and its constituent elements in their stable reference states. For a composition $A_x B_y \ldots$, the formation energy is defined as

$$E_{\text{formation}}(A_x B_y \ldots) = E(A_x B_y \ldots) - x E_A - y E_B - \ldots, \tag{50}$$

where $E_A$ denotes the energy of element A in its reference phase. A negative value with a sufficiently large magnitude of $E_{\text{formation}}$ indicates a more stable phase. Otherwise, the compound is more likely to decompose into the elements or into other phases. In materials design and screening, formation energies are used to construct phase diagrams, compute convex hulls, and evaluate decomposition energies.

In our comparative study, we evaluate the error distributions of formation energies for two training strategies. We present the results as cumulative distribution functions of the absolute error in Table 7. The Atomic Energy + Hessian curve largely coincides with the Hessian curve or shifts to the left, with consistent modest improvements in the middle to high error quantiles. This indicates that adding atomic energy supervision helps reduce the long tail of formation energy errors, which makes this metric of relative stability more robust.

### A.9 NVT MD SIMULATIONS

To further evaluate distilled MLFFs, we follow previous work Amin et al. (2025) and run 100 ps, constant temperature (NVT) MD simulations with systems from the Solvated Amino Acid subset. We choose 5 random structures from the test set as initial structures and perform Langevin dynamics at a temperature of 300K, a timestep of 1.0 $fs$, and a friction coefficient of 0.01 $fs^{-1}$, for 100,000 steps, corresponding to 100 ps. Consistent with (Fu et al., 2022), we use a metric of the maximum bond length deviation to measure stability. We keep track of stability through the bond lengths and say that a simulation becomes "unstable" at time $T$ if:

$$\max_{(i,j)\in\mathcal{B}} |(\|r_i(T) - r_j(T)\| - b_{i,j})| > \Delta, \tag{51}$$

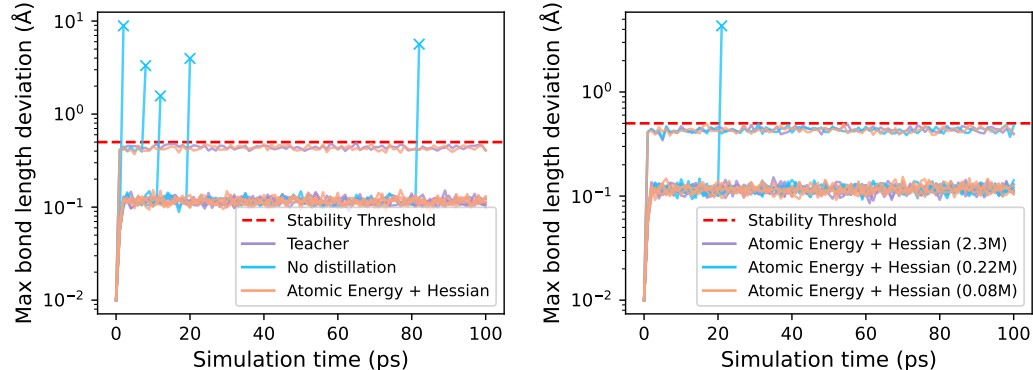

Figure 8: **Stability of NVT MD.** × denotes the point at which the simulation becomes unstable. The numbers in brackets in the right figure represent the number of model parameters. Our distilled models are generally more stable than the undistilled model.

where $\mathcal{B}$ is the set of all bonds, $i, j$ are the two endpoint atoms of the bond, and $b_{i,j}$ is the equilibrium bond length computed from the training dataset. Following (Fu et al., 2022), we set $\Delta = 0.5A$.

Experimental results from Amin et al. (2025) show that small undistilled models are unstable in MD, while Hessian distilled students markedly improve stability. Our results are shown in Figure 8 and findings are consistent: Atomic Energy + Hessian distillation maintains stable MD trajectories. We further compare students across sizes and observe generally robust stability. Only a single run with a smaller model exceeded the stability threshold. This indicates that adding Atomic Energy does not diminish the stability gains provided by Hessian distillation.

## A.10 The Use of Large Language Models (LLMs)

We used large language models (LLMs) to assist with drafting and polishing the manuscript text (improving clarity, grammar, and consistency of terminology). The LLM was not used to generate, analyze, or filter scientific results, and all LLM-assisted text was reviewed and edited by the authors.

