# OpenReview forum: "Learning Fast and Accurate Machine Learning Force Fields via Joint Atomic Energy and Energy Hessian Distillation"
_ICLR.cc/2026/Conference — Submitted to ICLR 2026_

### Official Review · Reviewer_UL2e · 2025-10-16

**Soundness:** 4
**Presentation:** 2
**Contribution:** 2
**Rating:** 4
**Confidence:** 4

**Summary:**

Leveraging the fact that the errors of energy, forces, and Hessians share the same Fourier coefficients weighted by some specific values, the authors propose a tailored distillation method for machine learning force fields. The proposed method significantly outperforms previous SOTA regarding energy MAE, and slightly reduces force MAE, with minimum overhead.

**Strengths:**

1. The paper is well-written;
2. The proposed method significantly outperforms previous SoTA regarding energy MAE with minimum overhead;
3. Intuitively, this method makes sense. The Hessian term only constraints the high-frequency components, leaving the low-frequency components, e.g., energy loosely constrained. Thus, the new loss penalty on the energy term can additionally constraint the low-frequency components;
4. They show that the distilled model is more stable via NVT molecular dynamics (MD) stability experiments, which is consistent with the benefit of better low-frequency constraints.

**Weaknesses:**

1. The biggest weakness of this work is the practicalness of the method. The method significantly improves the energy MAE, which is very good if we forget the application-side purpose. Nevertheless, as the title of the paper, it serves for the machine learning FORCE field. Force is very important for MLFFs, as it is the ultimate goal of MLFF itself, which are used for simulating molecular dynamics. Ideally, if the force prediction is highly robust, we do not even need the energy prediction, and some work did omit it [1], did not report it (see Table 3 in [2]) or significantly underperform others (see Table 2 in [3]). In fact, the energy prediction is used to derive the force by the negative gradient in most work, which can turn the model to a conservative force field, i.e., a good inductive bias. Thus, for MLFFs, one can view energy as a auxiliary term severing force prediction. It makes less sense that we predict energy well but only minor improvement for force prediction, even with some, although also minor, overhead.

2. The presentation is good in the Introduction section. However, it becomes less readable in Section 3. The authors poured out all the technical lemmata and theorems without an intuitive explanation of why and how we give each statement. I understand that there is the limitation of the main content. However, I recommend the authors indicate only the important theorems and lemmata in the main paper, leaving some technical lemmata to the appendix. Before giving a lemma, explain why we need it. Before and after giving a formal statement, explain the intuition and show the proof sketch. I recommend the authors refer to some well-written pure theoretical work, e.g., [4], to see how they make their paper highly readable (including the proof) while being theoretical heavy. That being said, my main concern is on weakness 1. If the authors can address it (decrease force MAE, or explain why performing poorly in force prediction is acceptable), I will raise my score and the readability can be tackled later in camera-ready.



[1] Hu W, Shuaibi M, Das A, et al. Forcenet: A graph neural network for large-scale quantum calculations[J]. arXiv preprint arXiv:2103.01436, 2021.

[2] Gasteiger J, Becker F, Günnemann S. Gemnet: Universal directional graph neural networks for molecules[J]. Advances in Neural Information Processing Systems, 2021, 34: 6790-6802.

[3] Musaelian A, Batzner S, Johansson A, et al. Learning local equivariant representations for large-scale atomistic dynamics[J]. Nature Communications, 2023, 14(1): 579.

[4] Raman V, Subedi U, Tewari A. Online learning with set-valued feedback[C]//The Thirty Seventh Annual Conference on Learning Theory. PMLR, 2024: 4381-4412.

**Questions:**

Please address weakness 1. Now that we have high-quality energy prediction, fine-tuning the force prediction based on it will help?

---

> ### Author Response · Authors · 2025-11-20
>
> Thank you for the thoughtful review and helpful comments.
>
> **1. Practical usage of our method**
>
> For molecular dynamics simulations, accurate model-predicted forces are required for a given system. However, energies are also an important quantity. For example, energy is crucial for stability assessment in MD simulations and for structure search applications [1]. We have added experiments on dihedral-angle scans and formation-energy prediction in Figure 5 and Figure 7. These two tasks demonstrate how the model can be used to find stable structures or conformations by energy.
>
> **2. Relation between forces and energies**
>
> In conservative models, forces are computed as the negative gradient of the potential energy. However, even if two models yield identical force predictions, their energies can still differ by an additive constant, which can yield large energy MAE. Similarly, low-frequency components of the energy error can lead to a large energy MAE but a small force MAE.Our improvements primarily address this issue, so the improvements in energy prediction have not resulted in significant improvements in force prediction.
>
> **3. Readability of the theoretical section**
>
> We acknowledge that Section 3 in the original submission was difficult to follow. In the revised version, we have already made a first round of edits to this section, adding explanations for each theorem to improve readability, and we will further refine the presentation.
>
> [1] Merchant, Amil, et al. "Scaling deep learning for materials discovery." Nature 624.7990 (2023): 80-85.

---

> > ### Comment · Reviewer_UL2e · 2025-11-21
> > **Thank you for your response**
> >
> > The authors' response and revision to the paper have addressed my concern. I have raised my score to a 6.
> >
> > I still have one more question for the authors: Do you find ``Machine-Learned Interatomic Potentials'' is a better title than MLFFs? If I understand correctly, the dihedral scan of the potential energy surface has nothing to do with the force part. Plus this application is the main use case that really shows off the strength of your method.

---

> > > ### Author Response · Authors · 2025-11-21
> > >
> > > Thank you again for the positive feedback. We are pleased that our revision has resolved your concern. We also appreciate your thoughtful suggestion about the title. *Interatomic Potentials* is indeed better than *Force Fields* for capturing the focus of our work, and we plan to adopt it in the next revision.

---

> > > > ### Comment · Reviewer_UL2e · 2025-11-21
> > > > **Thank you**
> > > >
> > > > My concerns have been all resolved. I will be willing to raise my score to between 6 and 8. Unfortunately, ICLR this year does not provide this option, so I reflect this both in my updated review and this comment. I wish good luck to the authors.

---

### Official Review · Reviewer_LTNS · 2025-10-30

**Soundness:** 3
**Presentation:** 3
**Contribution:** 3
**Rating:** 6
**Confidence:** 4

**Summary:**

This paper proposes a joint Atomic Energy–Energy Hessian Distillation framework for compressing large atomistic foundation models into smaller, faster specialized force-field models. The key idea is to augment Hessian-based distillation with an additional atomic-energy supervision term. Through a spectral (Fourier-domain) analysis, the authors argue that the energy, force, and Hessian objectives emphasize different frequency ranges of the energy-error spectrum ($\propto$ 1, $\omega^2$, $\omega^4$ respectively).
Combining atomic-energy (low-frequency) and Hessian (high-frequency) supervision therefore provides a more balanced constraint.

**Strengths:**

* Clear motivation and physical grounding: The work addresses a genuine need for distilling large foundation-model MLFFs into deployable sub-domain models.
* Clear derivation:
The Fourier-domain derivation explains how energy, force, and curvature supervision emphasize different frequency components of the error. This theoretical framing helps rationalize prior empirical results on Hessian distillation.
* Comprehensive experiments: Multiple teacher–student pairs (MACE-OFF, MACE-MP, eSEN → GemNet-dT, PaiNN, GemNet-T) and diverse datasets are covered.

**Weaknesses:**

1. Ambiguity around Figure 2: At first glance, I thought Figure 2 suggests that distilled students outperform the teacher, although the teacher’s bars are actually not shown. This can easily mislead readers. I also suggest the authors to explicitly plot teacher performance in Figure 2 for clearer comparison.
2. Use of real labels vs. pure distillation: The method always combines DFT-label supervision with distillation losses. Hence the apparent improvement over the teacher is unsurprising, since the student still learns from ground-truth data. I am very curious to see how the performance would change if the authors exclude DFT labels in the distillation.
3. Missing analysis of force-level distillation.
Since force corresponds to $\omega^2$ weighting—intermediate between energy (1) and Hessian ($\omega^4$)—one might expect that combining atomic energy + force distillation would capture most spectral benefits at significantly lower cost, avoiding second-order derivatives. The authors neither test nor justify why this more straightforward alternative is inferior. This omission weakens the claimed “minimal-cost yet spectrally complete” argument. In addition, if you really wish to include Hessian, why don't you simply include atomic energy + force + hessian in the distillation? Isn't this a straightforward way to do the distillation?
4. Lack of physical validation: Results focus on MAE metrics. Although there is one NVT simulation, the test is too simple and not enough.

**Questions:**

1. Did you test atomic energy + force distillation? If not, can you argue quantitatively that its spectral coverage is insufficient compared to atomic energy + Hessian?
2. In the Fourier proof, the zero/low-frequency modes of Hessian are shown to be not important, which is counterintuitive to me from the physics perspective. As low-frequency modes may still carry large group velocities and significantly impact the properties of the materials. I understand that this manuscript is not intended for a physics discussion, but this seems confusing to me.

---

> ### Author Response · Authors · 2025-11-20
>
> We appreciate your careful questions and feedback.
>
> **1. Impact of using ground-truth DFT labels**
>
> The idea of ​​combining true labels and distillation loss follows [1], which showed that combining the two is better than using true labels or distillation loss alone.
>
> Regarding the use of DFT labels, we extend our study on the Solvated Amino Acids dataset with GemNet-dT as the student model, and compare a setting where the loss is constructed only from teacher predictions to a setting where the ground-truth DFT labels (total energies and forces) and the teacher predictions (Hessians and atomic energies) are combined. The results indicate that the latter combination leads to better distillation accuracy. The corresponding experiments and discussion are reported in Appendix A.7.
>
> **2. On the use of teacher-predicted forces**
>
> Since the DFT labels already provide ground-truth forces, we use them rather than the teacher-predicted forces. Prior work [2] (Appendix, Table 15) explicitly tested replacing DFT forces with teacher-predicted forces and observed that this substitution leads to a drop in student accuracy.
>
> We aim to start from DFT energies and forces as the primary supervision terms, and then extract additional useful information from the teacher model to further improve the distilled student model. Atomic energies and Hessians do not have DFT labels and can be obtained from the teacher. Therefore, our distillation scheme leverages DFT energies and forces, as well as the teacher’s atomic energies and Hessians.
>
> **3. Hessian and low-frequency information**
>
> We agree with the reviewer on the crucial physical role of low-frequency components. To clarify the apparent tension with physical intuition: our statement that Hessian supervision de-emphasizes zero/low-frequency error refers to the spectral weighting in the learning objective (i.e., how different frequencies of the error field are penalized), not to the physical importance of low-frequency phonon modes. To better capture the low-frequency components, we additionally introduce atomic energies as a complementary supervision term. We have revised Section 3.1 to make this point clearer.
>
> **4. Lack of physical validation**
>
> We have added two tasks: dihedral-angle scans and formation-energy. The corresponding results are reported in Figure 5 and Figure 7. These experiments show that our distillation method produces predictions that are closer to the DFT reference than the method of [2].
>
> **5. Clarification of Figure 2**
>
> The accuracy of the teacher model is reported in Appendix Tables 2 and 3. In the revised paper, we have added the teacher’s accuracy to Figure 2 for better comparison.
>
> [1] Hinton, Geoffrey, Oriol Vinyals, and Jeff Dean. "Distilling the knowledge in a neural network." arXiv preprint arXiv:1503.02531 (2015).
>
> [2] Amin, Ishan, Sanjeev Raja, and Aditi Krishnapriyan. "Towards fast, specialized machine learning force fields: Distilling foundation models via energy hessians." 13th International Conference on Learning Representations, ICLR 2025.

---

### Official Review · Reviewer_rCXa · 2025-10-31

**Soundness:** 3
**Presentation:** 3
**Contribution:** 3
**Rating:** 2
**Confidence:** 3

**Summary:**

The paper propses a knowledge distillation method for foundation models which compute/estimate atomic forces and energies of molecules/solids. In order to speed up inference (for example to be able to conduct long time MD simulations), specialized smaller systems can be extracted from such large foundation models. These smaller models are then trained in a student teacher setting. Ideally these smaller models keep the accuracy of the large model (for their specific task) while yielding faster inference by several orders of magnitude.

Concretely, the paper augments previous approaches (which only penalize the deviation in the Hessian of the student/teacher models) by introducing an additional term that penalizes the deviation in atomic energies. It is argued that the Hessian only controls higher frequency components and therefore the additional term serves to control also the low frequency error.

The experiments show improvements in training speed over a purely Hessian regularization with the computational overhead due to the additional penalty being almost negligible. In terms of accuracy, the trained model is comparable to the purely Hessian approach.

**Strengths:**

The paper tackles an important topic. Combining large foundation models with knowledge distillation for specialized tasks in computational chemistry is a beautiful idea.

The main observation of the paper is that the Hessian does not constrain the low Fourier modes and therefore it needs to be augmented. The paper presents some theoretical justification for this fact.

**Weaknesses:**

The results on Fourier modes are trivial and obvious. If I understand correctly, the main practical contribution is to simply add another term to the optimization objective. Overall I have doubts that the conceptual novelty of the paper is sufficient for ICLR.

**Questions:**

Why do you assume that the error function is periodic? Can you justify this?

I have doubts that Hessian regularization is a viable option for simulating large particle numbers, due to the quadratic complexity. Is there any way to address this?

---

> ### Author Response · Authors · 2025-11-20
>
> Thank you for thoughtful questions.
>
> **1. Novelty**
>
> Our work starts by revisiting the role of the Hessian through a Fourier-based analysis, showing that it predominantly captures high-frequency components of the energy landscape. This viewpoint helps understand the mechanism behind the Hessian distillation approach of Amin et al. [1] more deeply. On top of this, we argue that low-frequency information is equally essential for a stable and accurate distillation process, and that such low-frequency components can be naturally obtained by aligning atomic energies between the teacher and student models.
>
> DFT total energies already serve as supervision in both the Hessian distillation method [1] and our method. At the same time, atomic energies were not used in the prior Hessian distillation method [1]. We perform a Fourier analysis of the Hessian and atomic energies and show that they contain high- and low-frequency information from teacher. So we further introduce atomic energies from the teacher model. From a distributional perspective, atomic energies provide finer-grained supervision that reshapes the student’s atomic energy landscape beyond what total energies and Hessians alone can offer. We conduct comprehensive experiments and demonstrate that incorporating the Hessian and atomic energies further improves the student model’s predictions.
>
> **2. Periodicity of the loss function**
>
> The periodicity of the loss function follows from the periodicity of the potential energy, which arises from the periodic boundary conditions of the physical system. For crystalline systems, such periodic boundaries are intrinsic. For non-periodic molecular systems, it can be embedded in a sufficiently large periodic box to avoid cross-boundary interactions.
>
> Moreover, the chemically relevant configuration space we care about is within a finite region. When evaluating the error, it is only necessary to integrate over a finite space. Under this restriction, adopting periodic boundary conditions is a reasonable approximation that avoids dealing with boundary terms and does not affect results within the region of interest. We clarify this point in Section 3 of the main text.
>
> **3. On the quadratic complexity of the Hessian**
>
> The Hessian is only used during distillation to improve the accuracy of the student model. Once the student has been trained, inference relies solely on energy and forces, and does not require calculating Hessian, so there is no additional cost in the inference process.
>
> In the training stage, the Hessian of the teacher model for each sample is computed only once. For the student model, it is not necessary to compute the full Hessian matrix at every update. When calculating the loss, we approximate the Hessian term by computing a constant number of rows. Our experiments show that sampling only a fixed number of rows per iteration is sufficient to maintain high accuracy for the student model, thereby keeping the effective computational complexity linear rather than quadratic.
>
> [1] Amin, Ishan, Sanjeev Raja, and Aditi Krishnapriyan. "Towards fast, specialized machine learning force fields: Distilling foundation models via energy hessians." 13th International Conference on Learning Representations, ICLR 2025.

---

> > ### Comment · Reviewer_rCXa · 2025-11-21
> >
> > Thank you very much for your comments!
> > I still feel that the theoretical Fourier analysis is trivial and the main contribution is to add another penalty term. To me this is not sufficiently novel to mandate publication in ICLR and therefore I am keeping my score.

---

> > > ### Author Response · Authors · 2025-12-03
> > >
> > > Thank you again for the feedback. The main innovations of our work are the proposal of a joint distillation loss that combines Hessian-based high-frequency constraints with atomic-energy low-frequency supervision, and the thorough experimental validation across settings and tasks. The theoretical part provides an intuitive spectral explanation for why this combination works. Regarding the concern about just adding another penalty term, we respectfully note that in distillation the central contribution lies in designing the loss term, and our formulation is consistent with this objective.

---

### Official Review · Reviewer_uHZU · 2025-11-10

**Soundness:** 2
**Presentation:** 3
**Contribution:** 2
**Rating:** 2
**Confidence:** 4

**Summary:**

This paper looks at knowledge distillation for machine learning force fields. The authors look at the previously proposed Hessian distillation and the Fourier coefficients for energy, forces, and the Hessians. They then add an additional atomic energy term for supervision during distillation. They demonstrate results by distilling from models trained on the SPICE, MPTrj, and OMat24 datasets.

**Strengths:**

- The frequency analysis of the errors of the different terms is interesting, though needs to be more rigorously quantified

**Weaknesses:**

- The work seems very incremental compared to prior work on Hessian distillation, as the main contribution is adding an additional energy term during distillation supervision. The primary results that the authors show are improvements in energy and force MAE on the SPICE and MPTrj datasets. The paper would benefit from having new examples and downstream use cases to more rigorously quantify what improvements they may be seeing. It would also be important to go beyond energy and force MAEs, as these do not always necessarily correlate to downstream performance. Additionally, it would be helpful to see if such a procedure enables new accuracy and timescales that were not possible before.

- The authors are listing specific lemmas and theorems in Section 3.1, but it is not fully clear how these follow: the paper would benefit from providing more rigorous proofs for each statement

**Questions:**

- How did the authors obtain equation 16, as well as Theorem 3.4?

- What models are being used in Figure 4?

- Can you provide additional downstream use cases to more rigorously show the benefits of adding the extra energy term during the distillation procedure? What does it enable that was not possible before?

---

> ### Author Response · Authors · 2025-11-20
>
> Thank you for thoughtful comments and suggestions.
>
> **1. Downstream application cases**
>
> We agree that reductions in the MAE of energies and forces do not always imply improvements on downstream tasks. Here, we consider two typical downstream applications: dihedral-angle scans and formation-energy prediction.
>
> For the dihedral-angle scans task, our distillation method (Atomic Energy + Hessian) aligns more closely with DFT than the model obtained by Hessian-only distillation. The corresponding experimental results are reported in Figure 5.
>
> For formation energies, we plot the cumulative distribution of absolute errors in Figure 7. The results show that Atomic Energy + Hessian not only reduces total-energy MAE but also yields better predictions for formation-energy.
>
> **2. Derivations of lemmas and theorems**
>
> Due to space limitations in the main text, the detailed derivations of each lemma and theorem are provided in Appendix A.2. Specifically, the proof of Eq. (16) is shown in Line 824 to Line 840 and the proof of Theorem 3.4 is displayed in Line 841 to Line 858.
>
> **3. Contributions of this work**
>
> We analyze the Hessian from a Fourier perspective and point out that the Hessian encodes high-frequency information. This theoretical analysis helps clarify the mechanism of Hessian distillation method [1]. Building on this analysis, we argue that low-frequency information is also crucial during distillation and can be represented by atomic energies.
>
> Atomic energies were not exploited in the Hessian distillation work [1]. Since the DFT total energies were already used as labels (in both Hessian distillation method and ours), the fact that adding atomic energies from teacher model can improve the accuracy of student model is nontrivial. In this context, the role of the teacher's atomic energy is to enable the student model to learn the teacher model's atomic energy decomposition. In the experimental section, we demonstrate the effectiveness of incorporating atomic energies.
>
> **4. Clarification on the model used in Figure 4**
>
> In Figure 4, the student model is GemNet-dT, and the evaluation is conducted on the Solvated Amino Acids dataset. We have revised the figure caption in the main text to state this more clearly.
>
> [1] Amin, Ishan, Sanjeev Raja, and Aditi Krishnapriyan. "Towards fast, specialized machine learning force fields: Distilling foundation models via energy hessians." 13th International Conference on Learning Representations, ICLR 2025.

---

### Author Response · Authors · 2025-11-20

Dear Reviewers,

Thank you for your review and constructive feedback. We have made the following updates in the revised manuscript. All changes are highlighted in blue for easy reference.

**Added downstream evaluations:**

- Include dihedral-angle scan experiments to demonstrate improvements in conformational stability assessment and potential energy surface characterization.

- Add formation energy prediction experiments to assess practical value for structure search and materials-related tasks.

**Improved theoretical analysis and readability:**

- Substantially revise Section 3 for clarity, adding transitions between key theorems/lemmas to enhance readability.

**Clarified previously ambiguous descriptions:**

- Revise and clarify wording and figures that were potentially ambiguous in response to reviewer comments.

We appreciate your constructive suggestions. These revisions substantially strengthen the experimental completeness and the readability of the theory. We welcome further feedback and will continue to improve the manuscript.

---

### Author Response · Authors · 2025-12-03
**Summary for the Area Chair**

Dear Area Chair,

Our paper proposes a distillation method that introduces atomic‑energy supervision to state-of-the-art Hessian‑based distillation, achieving improved energy‑prediction accuracy with minimal overhead compared to Hessian distillation, and demonstrating effectiveness across multiple downstream tasks.

We thank all reviewers for their thoughtful comments and suggestions. Below is a concise summary of the main reviews and our responses.

**Strengths**

- **Clear motivation and relevance:** Reviewers rCXa and LTNS agreed that distilling large atomistic foundation models into efficient specialized models addresses an important and meaningful problem.

- **Insightful spectral interpretation:** Reviewers uHZU, rCXa, and LTNS found the Fourier‑domain perspective helpful for understanding how Hessian and atomic‑energy supervision provide complementary high‑ and low‑frequency constraints.

- **Strong empirical improvements with minimal overhead:** Reviewers uHZU and UL2e highlighted that the method delivers clear gains in energy accuracy and improved MD stability while adding negligible computational cost.

**Concerns & Responses**

- **Downstream validity/insufficient physical validation/limited force improvement (uHZU, LTNS, UL2e):** We added dihedral-angle scans and formation-energy prediction, showing PES closer to DFT and better formation-energy distributions. Using spectral weighting, we explain why energy gains may not translate to lower force MAE, while still offering practical value.

- **Novelty (rCXa):** Our method introduces an atomic-energy supervision term, motivated by a complementary frequency-domain view, and its necessity is supported by downstream results.

- **Specific issues:** Readability/proof completeness (uHZU, UL2e): The theory section was rewritten with full proofs in the appendix. Figure 4 (uHZU): Clarified as GemNet-dT on Solvated Amino Acids. Periodicity assumption (rCXa): Justified by PBC and finite-domain approximation without affecting conclusions. Quadratic complexity (rCXa): The teacher Hessian is computed once; student uses row-sampling to keep training cost linear.

During the rebuttal period, we addressed reviewer UL2e’s concerns, and he indicated that he was willing to raise the score from 4 to between 6 and 8 (confidence 4). Reviewer LTNS, whose confidence is also 4, gave us a score of 6. He did not manage to reply before the OpenReview bug occurred, but he understood our contributions, and we further clarified several of his questions. Reviewer uHZU raised concerns similar to UL2e’s, but he also did not manage to reply. Reviewer rCXa did not fully grasp the essence of our work. He believes that our theory is too simple and that our contribution is merely adding an extra loss function term. However, the theoretical part is intended only to provide intuition. Our main focus is on the study of distillation methods, and distillation methods are fundamentally about the design of loss functions. While adding an extra loss function term may seem straightforward at the implementation level, this formulation ensures that the distilled model preserves both the high‑frequency and low‑frequency information of the potential energy surface. We conducted extensive experiments to validate the effectiveness of this distillation method, and its advantages are further reflected across several downstream tasks.

Overall, we believe this work makes a meaningful contribution. We thank the reviewers for their thoughtful feedback and have addressed their concerns to further improve the paper. We sincerely appreciate the Area Chair and the reviewers for their time and consideration.

Sincerely,

The Authors

---

### Meta-Review · Area_Chair_RXrB · 2026-01-06

**Summary:**

This paper improves the existing distillation framework for compressing large atomistic foundation models into smaller, faster specialized force-field models. The key idea is to augment Hessian-based distillation with an additional atomic-energy supervision term. The reviewers raise concerns that the main contribution of the paper is incremental, consisting primarily of adding an additional term to the distillation supervision. While this modification leads to some improvement in energy prediction, the gains in force prediction are limited. Moreover, despite long-time molecular dynamics being a central motivation of the work, the paper does not provide compelling validation on long MD simulations, which weakens the practical impact of the proposed approach.

**Reviewer Concerns:**

The rebuttal partially addresses the reviewers’ concerns. The authors reorganize the presentation of the theoretical results, which helps clarify the frequency-based intuition behind the method. However, the core concerns regarding the incremental nature of the contribution, the limited improvement in force prediction, and the lack of compelling long-time MD validation remain largely outstanding.

**Reviewer Scores:**

Reviewer uHZU is likely to maintain a score of 2, as the contribution remains incremental and the additional energy-related downstream results are still not fully convincing.

Reviewer rCXa has indicated they would keep a score of 2, viewing the Fourier-based theoretical analysis as relatively straightforward and considering the main contribution to be the introduction of an additional penalty term.

Reviewer LTNS is likely to keep their original score of 6, as the added physical validation remains limited (still some energy calculation) and the distillation results without original labels perform worse.

Reviewer UL2e has indicated they would increase their score from 4 to 6.

---

### Decision · Program_Chairs · 2026-01-26

Reject